# FactGuard: Agentic Video Misinformation Detection via Reinforcement Learning

**Zehao Li** [* 1 2]  **Hongwei Yu** [* 3]  **Hao Jiang** [1 2]  **Qiang Sheng** [1 2]  **Yilong Xu** [1 2]
**Baolong Bi** [1 2]  **Yang Li** [1 2]  **Zhenlong Yuan** [1 2]  **Yujun Cai** [4]  **Zhaoqi Wang** [1 2]

## Abstract

Multimodal large language models (MLLMs) have substantially advanced video misinformation detection through unified multimodal reasoning, but they often rely on fixed-depth inference and place excessive trust in internally generated assumptions, particularly in scenarios where critical evidence is sparse, fragmented, or requires external verification. To address these limitations, we propose **FactGuard**, an agentic framework for video misinformation detection that formulates verification as an iterative reasoning process built upon MLLMs. FactGuard explicitly assesses task ambiguity and selectively invokes external tools to acquire critical evidence, enabling progressive refinement of reasoning trajectories. To further strengthen this capability, we introduce a two-stage training strategy that combines domain-specific agentic supervised fine-tuning with decision-aware reinforcement learning to optimize tool usage and calibrate risk-sensitive decision making. Extensive experiments on FakeSV, FakeTT, and FakeVV demonstrate FactGuard's state-of-the-art performance and validate its excellent robustness and generalization capacity.

## 1. Introduction

The rapid growth of large-scale online content-sharing platforms, such as TikTok, has accelerated the spread of information (Bu et al., 2023). This accessibility creates a fundamental asymmetry between rapid misinformation propagation and delayed manual verification, allowing misleading

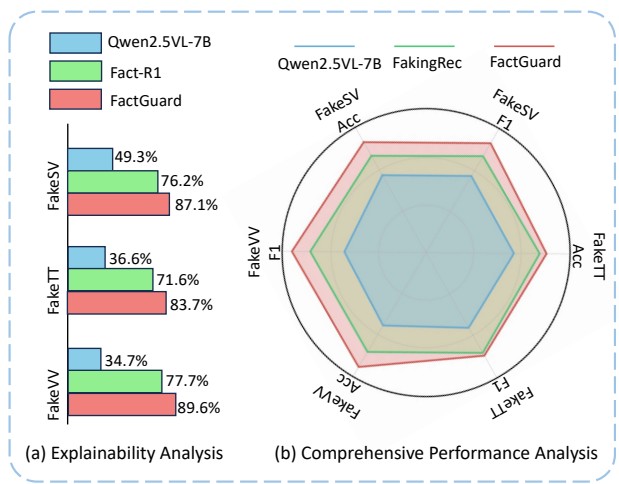

Figure 1. Comparison of video misinformation detection methods and our proposed **FactGuard** in terms of (a) explainability analysis and (b) comprehensive performance analysis.

content to influence public discourse beyond the limits of human intervention. Consequently, post-hoc fact-checking becomes increasingly inadequate, highlighting the need for accurate and timely automated misinformation detection (Sheng et al., 2025; Shang et al., 2021).

Among various content modalities, videos have emerged as a dominant and particularly challenging medium for misinformation dissemination due to their rich temporal dynamics and inherent multimodal complexity. In recent years, increasing attention has therefore been devoted to video misinformation detection, yielding encouraging progress. However, most existing approaches (Qi et al., 2023a; Bu et al., 2024; Qi et al., 2023b) rely on task-specific discriminative models and lack the general understanding and reasoning capabilities, which are required for handling various verification needs in open-world scenarios.

The rapid advancement of large-scale multimodal models has significantly improved multimodal content understanding and enabled reasoning-based approaches for misinformation verification. However, most existing methods remain constrained by a single-pass inference paradigm, lacking explicit mechanisms for uncertainty awareness and targeted

*Equal contribution [1]Institute of Computing Technology, Chinese Academy of Sciences, Beijing, China [2]University of Chinese Academy of Sciences, Beijing, China [3]University of Science and Technology Beijing, Beijing, China [4]The University of Queensland, Brisbane, Australia. Correspondence to: Hao Jiang <jianghao@ict.ac.cn>.

*Proceedings of the 43rd International Conference on Machine Learning*, Seoul, South Korea. PMLR 306, 2026. Copyright 2026 by the author(s).

evidence acquisition. In ambiguous or weakly verifiable scenarios, models therefore tend to rely on internally generated assumptions rather than grounding their reasoning in modality-specific or external evidence. As illustrated in Figure 3(a), this behavior results in **cross-modal hallucinations**, where the model fabricates or misattributes visual or textual evidence that is not present in the input, and consequently leads to **erroneous verification outcomes**, namely confident but incorrect judgments about the factuality of the underlying claim. Consequently, reliable misinformation verification requires a system that can (1) identify when available information is insufficient, (2) selectively acquire targeted external evidence, and (3) iteratively refine its judgments based on new observations.

Motivated by this insight, we propose **FactGuard**, an end-to-end agentic framework for video misinformation detection built on multimodal large language models (MLLMs). FactGuard formulates verification as an uncertainty-aware, iterative decision-making process, selectively invoking external tools when necessary to support reliable verification. Specifically, FactGuard introduces an agentic reasoning pipeline that integrates multimodal deliberation with explicit tool invocation and evidence-driven refinement. When the available information is insufficient for confident verification, the model adaptively selects external tools based on its reasoning state, including *FactProbe* for external knowledge verification and *ClipScout* for targeted visual evidence inspection, to supplement textual and visual inputs. The acquired evidence is incorporated into subsequent reasoning stages, enabling more informed and reliable verification outcomes. To further strengthen this capability, we construct a multimodal agentic Chain-of-Thought dataset for misinformation detection and perform supervised fine-tuning to establish structured reasoning and tool-invocation behaviors. We additionally incorporate a decision-aware reinforcement learning strategy that reinforces evidence-grounded reasoning while explicitly modeling tool usage, asymmetric error costs, and decision preferences under uncertainty. Together, these design choices enable FactGuard to move beyond existing approaches, establishing a general and interpretable verification framework, as shown in Figure 1.

Our main contributions are as follows:

- We propose **FactGuard**, an agentic multimodal framework that formulates video misinformation detection as an iterative verification process with self-reflective reasoning and selective evidence acquisition.

- We construct a multimodal agentic Chain-of-Thought dataset for misinformation detection and perform targeted CoT-based supervised fine-tuning to inject domain-specific reasoning behaviors into MLLMs.

- We develop a decision-aware reinforcement learning strat-

egy that explicitly models tool usage, asymmetric error costs, and decision-making preferences, leading to more calibrated and reliable verification outcomes.

- Extensive experiments on FakeSV, FakeTT, and FakeVV datasets demonstrate that FactGuard achieves state-of-the-art performance and consistently outperforms existing methods by a significant margin.

## 2. Related Work

### 2.1. Misinformation Detection

Early studies on multimodal misinformation detection mainly focused on the image–text setting, where static visual content can be directly aligned with textual claims. These approaches (Wang et al., 2018; Qian et al., 2021; Wang et al., 2023) model cross-modal interactions via attention, feature fusion, or contrastive learning to capture semantic consistency between images and text, achieving promising performance on social media benchmarks.

In recent years, research on misinformation detection has moved beyond static images to the video domain, driven by the growing prevalence of video-based misinformation on social media platforms. Existing approaches (Qi et al., 2021; Wang et al., 2025b) typically focus on exploiting the rich multimodal signals embedded in videos by jointly modeling visual, acoustic, and textual information. Some studies (Mittal et al., 2020; Xu et al., 2025) enhance detection performance by integrating linguistic patterns with emotional or prosodic cues from audio, while others (Wang et al., 2025a; McCrae et al., 2022) explicitly model cross-modal inconsistencies among video frames, audio streams, and subtitles. In addition, prior work (Qureshi et al., 2021) has explored cross-channel watermarking for manipulation detection, as well as multimodal fusion frameworks that combine topic representations with keyframe-level visual features. More recent efforts (Qi et al., 2023b; Gong et al., 2025; Qi et al., 2023a) further incorporate social context or neighborhood structures to capture relational dependencies among multimodal samples.

### 2.2. Multimodal LLMs Reasoning

Recent advances in multimodal large models (MLMs) have significantly improved the ability to jointly understand and reason over visual, textual, and auditory inputs. Models such as GPT-4V (Yang et al., 2023), LLaVA (Guo et al., 2024; Li et al., 2023; Zhang et al., 2025b; Lin et al., 2024), and Qwen-VL (Bai et al., 2023; Wang et al., 2024a; Yang et al., 2025) demonstrate strong generalization across diverse multimodal tasks, enabling more expressive reasoning beyond traditional feature-level fusion.

Building on these models, recent work has increasingly

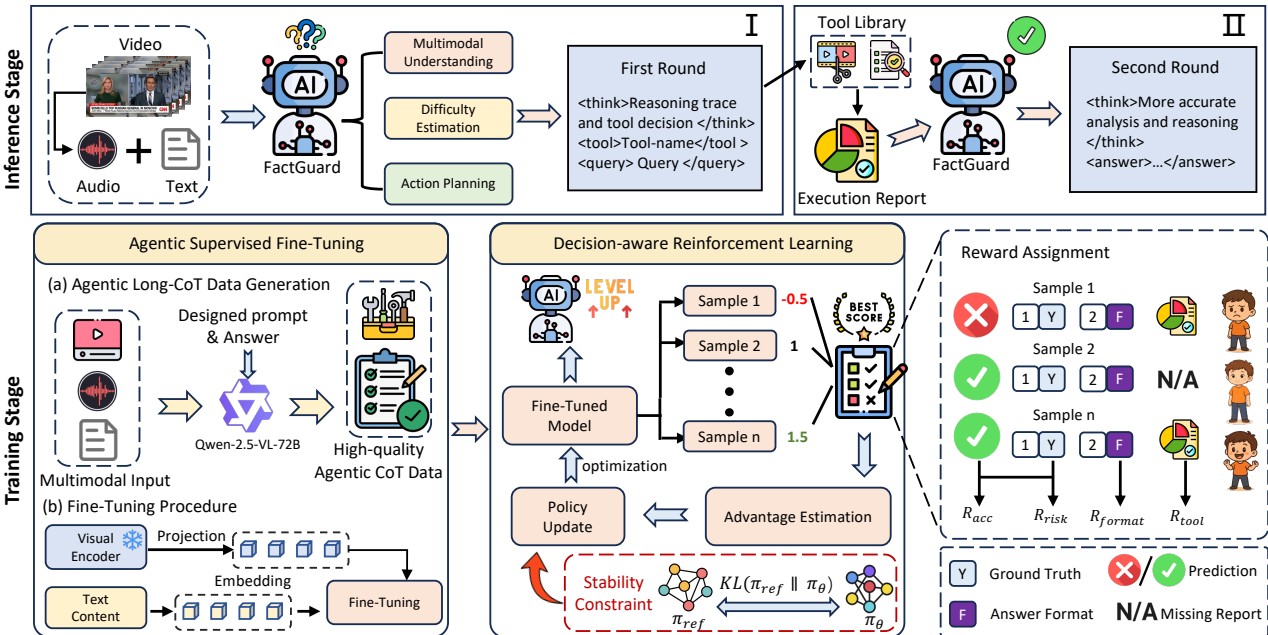

*Figure 2.* **Pipeline of FactGuard.** The upper part illustrates the inference-time agentic verification process, where FactGuard assesses uncertainty based on the ambiguity of the input and selectively invokes external tools to acquire additional evidence before refining its reasoning and producing a final decision. The lower part depicts the training pipeline, which combines supervised fine-tuning with decision-aware reinforcement learning to reinforce structured reasoning, calibrated tool usage, and risk-sensitive verification behavior.

explored multimodal reasoning, including visual grounding (Alayrac et al., 2022), reasoning process prompting (Wei et al., 2022), and explanation generation (Cheng et al., 2023). Recent studies on video hallucination further emphasize the importance of fine-grained spatial-temporal grounding for reliable video reasoning (Luo et al., 2025), which is particularly relevant to misinformation verification where misleading cues may appear in specific frames or temporal segments. By explicitly modeling intermediate reasoning steps, these methods improve both interpretability and performance on complex perceptual tasks. Meanwhile, recent studies on context-faithful alignment (Bi et al., 2024) highlight the importance of encouraging language models to ground their responses in provided evidence rather than relying on unsupported parametric knowledge. Related work further investigates how models balance parametric knowledge and external contexts (Bi et al., 2025b), which is closely related to misinformation verification where models must decide whether to trust their internal knowledge or newly acquired multimodal evidence.

Reinforcement learning–based post-training has further enhanced model capabilities, as demonstrated by OpenAI-o1 (Jaech et al., 2024b) and DeepSeek-R1 (Guo et al., 2025). Recent studies also show that designed rewards and rubrics can guide models toward more effective exploration and reasoning behaviors (Bi et al., 2025a). In misinformation detection, early efforts such as Fact-R1 (Zhang et al., 2025a) demonstrate the promise of combining large pretrained

models with domain-specific fine-tuning and reinforcement learning, but they still primarily follow a single-pass verification paradigm without adaptive evidence acquisition.

### 2.3. Tool-Augmented Agentic System

Recent works (Zheng et al., 2025; Cui et al., 2025) have increasingly explored augmenting multimodal models with external tools to support more complex reasoning and adaptive information access. Early studies (Li et al., 2025; Sun et al., 2025) introduce tools as auxiliary sources of visual evidence, enabling models to ground intermediate reasoning steps in external perception signals. Subsequent efforts (Liu et al., 2024; Zhao et al., 2025) investigate how tool usage can be learned rather than manually specified, through supervision or reinforcement signals that align tool invocation with task objectives. More recent agentic frameworks further extend this paradigm to domain-specific visual understanding by dynamically invoking external tools and optimizing tool efficiency through reinforcement learning (Tan et al., 2026; Yuan et al., 2026).

However, tool-augmented and agentic reasoning paradigms remain largely unexplored in video misinformation detection, which is still dominated by single-pass verification without iterative evidence acquisition or adaptive decision-making. In contrast, our work introduces an agentic, tool-augmented verification framework that enables iterative and evidence-driven reasoning.

# 3. Methodology

## 3.1. Overview

As illustrated in Figure 2, FactGuard formulates video misinformation verification as an agentic, iterative decision-making process that explicitly integrates multimodal reasoning, evidence-guided action, and outcome-aware optimization. In the following sections, we first introduce the agentic formulation of FactGuard and its two-stage inference process (Section 3.2), and then describe the evidence-guided action module that governs tool invocation and information acquisition, including *FactProbe* for external knowledge retrieval and *ClipScout* for targeted video-clip inspection (Section 3.3). We subsequently detail the training pipeline, including agentic Chain-of-Thought supervised fine-tuning (Section 3.4), and decision-aware reinforcement learning with structured rewards (Section 3.5).

## 3.2. Problem Formulation

Each input sample is represented as a triplet $n = (n^{\text{vid}}, n^{\text{aud}}, n^{\text{txt}})$, where $n^{\text{vid}}$ denotes the news video content, $n^{\text{aud}}$ is the speech-to-text transcript extracted from the audio stream, and $n^{\text{txt}}$ refers to textual metadata such as titles and keywords.

Upon receiving the full multimodal input $n$, the agent applies an agent policy $\mathcal{A}_\theta$ to perform an initial chain-of-thought reasoning pass for assessing the difficulty and ambiguity of the case. The first-stage inference process is abstracted as:

$$(r_t,\ a_t) = \mathcal{A}_\theta^{(1)}(n,\ s_t), \qquad (1)$$

where $r_t$ denotes the reasoning trajectory generated at step $t$, and $a_t$ is the agent's action decision indicating whether to invoke an external tool. The decision is conditioned on the initial observation $n$ together with the agent state $s_t$, which captures its current belief and uncertainty.

If a tool is executed and returns an observation $o_t$, the agent incorporates this evidence and proceeds to a second-stage, evidence-augmented reasoning process. This refinement stage jointly considers both the original multimodal input and the tool feedback, and is formalized as:

$$(r_{t+1},\ \hat{y}) = \mathcal{A}_\theta^{(2)}(n,\ o_t,\ s_{t+1}), \qquad (2)$$

where $r_{t+1}$ denotes the refined reasoning trajectory and $\hat{y}$ is the final verdict. The second-stage reasoning conditions on both the original input $n$ and the tool-provided evidence $o_t$, enabling the agent to update its decision based on external verification signals. This agentic formulation promotes cautious, evidence-driven verification behaviour, which is essential for robust misinformation detection under ambiguous and information-scarce real-world conditions.

## 3.3. Evidence-Guided Action Module

When the first-stage reasoning identifies missing context or unresolved uncertainty, FactGuard issues an evidence-guided action to acquire supplementary information. Rather than relying on tools by default, tool invocation is treated as a deliberate verification decision that is triggered only when internal reasoning is deemed insufficient. In practice, FactGuard supports multiple evidence acquisition tools, including **FactProbe** and **ClipScout**.

Formally, given an action decision at step $t$, the agent invokes a tool operator $\mathcal{T}$, which executes the corresponding query and returns an observation:

$$o_t = \mathcal{T}(\kappa_t, \psi_t), \qquad (3)$$

where $\kappa_t$ specifies the selected tool identity and $\psi_t$ denotes the associated query parameters.

**External Knowledge Retrieval (FactProbe).** Certain misinformation claims cannot be resolved solely from visual or audio cues, particularly those involving real-world events, historical facts, or temporal assertions. To address this limitation, FactGuard employs **FactProbe**, a retrieval-augmented knowledge access module that formulates structured factual queries based on the model's current reasoning state and retrieves evidence from web-based sources. Retrieved results are filtered using reliability heuristics and summarized into a concise textual report, which is incorporated as auxiliary evidence in the refinement stage to supplement incomplete or uncertain textual information and support more reliable verification.

**Video-Clip Temporal Inspection (ClipScout).** Misinformation videos often exhibit long durations with substantial visual redundancy, while only a small number of localized temporal segments contain decisive evidence. ClipScout is designed to enable targeted inspection of such evidence-bearing intervals by selectively sampling representative frames from queried time spans and aggregating them into a compact visual summary. This focused visual evidence supports evidence-oriented reasoning by guiding the model's attention toward salient events. Through repeated agentic interaction and training, the model learns to efficiently localize and ground critical visual cues without exhaustively processing the entire video.

## 3.4. Agentic Supervised Fine-Tuning

To address the limited reasoning horizon and underdeveloped tool-usage behaviors of existing multimodal models in misinformation detection, we construct a misinformation-oriented agentic Chain-of-Thought (CoT) dataset that provides structured demonstrations of multi-round reasoning, tool selection intent, and evidence-grounded reflection.

Each annotated trajectory records how the agent analyzes multimodal content, determines when external evidence is required, and incorporates retrieved evidence to refine subsequent reasoning and the final judgment. This reframes misinformation detection from a one-shot classification task into an agentic, self-regulated decision-making process.

We then perform chain-of-thought supervised fine-tuning to align the model with these agentic reasoning behaviors. Given an input $n$ and its annotated agent trajectory $\tau = (r_t, a_t, o_t, r_{t+1}, \hat{y})$, the model is trained to maximize the likelihood of the full reasoning and action sequence:

$$\mathcal{L}_{\text{SFT}} = -\mathbb{E}_{(n,\tau)\sim\mathcal{D}}\big[\log p_\theta(\tau \mid n)\big]. \quad (4)$$

This supervision encourages the model to internalize longer reasoning chains, explicit uncertainty awareness, and evidence-seeking preferences in ambiguous cases. As a result, FactGuard avoids premature or over-confident conclusions and develops verification-oriented reasoning behaviors that are critical for real-world misinformation detection.

### 3.5. Decision-Aware Reinforcement Learning

To further calibrate the verification policy under uncertainty, we introduce a decision-aware reinforcement learning stage.

**Group Relative Policy Optimization (GRPO).** We adopt GRPO as a critic-free reinforcement learning paradigm that operates at the level of full agent trajectories. For each input instance, the policy generates a group of candidate trajectories, each reflecting different reasoning depth, tool-invocation behaviour, and final prediction outcomes. GRPO performs groupwise comparison over these trajectories and updates the policy according to their relative verification quality. The policy network $\pi_\theta$ is optimized by an importance-weighted objective with KL regularization against a frozen reference model:

$$\mathcal{L}_{GRPO}(\theta) = \mathbb{E}_{q\sim P(Q),\{o_i\}_{i=1}^{G}\sim\pi_{\theta_{\text{old}}}(O|q)}$$
$$\left[\sum_{i=1}^{G}\frac{\pi_\theta(o_i|q)}{\pi_{\theta_{\text{old}}}(o_i|q)}\cdot A_i - \beta\mathbb{D}_{KL}(\pi_\theta||\pi_{ref})\right], \quad (5)$$

$$\mathbb{D}_{KL}(\pi_\theta||\pi_{ref}) = \frac{\pi_{ref}(o|q)}{\pi_\theta(o|q)} - \log\frac{\pi_{ref}(o|q)}{\pi_\theta(o|q)} - 1, \quad (6)$$

where $\beta$ controls the trade-off between exploration and stability, $\pi_{\theta_{\text{old}}}$ denotes the rollout policy used for trajectory sampling, and $\pi_{\text{ref}}$ is a frozen reference model that stabilizes policy updates. The KL term serves as a pointwise surrogate of the KL divergence for efficient policy regularization. Each trajectory $\tau_i$ is assigned a task-specific reward based on verification reliability, including the correctness of the decision and the quality of its reasoning and tool usage. To enable stable comparison within each trajectory group,

we compute a normalized advantage:

$$A_i = \frac{R(\tau_i) - \text{mean}(\{R(\tau_j)\})}{\text{std}(\{R(\tau_j)\})}, \quad (7)$$

which reduces reward-scale variance and emphasizes trajectories exhibiting stronger evidence-grounded agent behaviours relative to their peers.

**Reward Design.** To ensure that reinforcement learning optimizes verification-oriented decision behavior rather than merely maximizing label accuracy, we design a gated reward function that jointly models decision correctness, tool usage, and risk sensitivity for multimodal misinformation detection. We first define the overall trajectory reward and then describe each component in turn. Specifically, for each agent trajectory $\tau$, the total reward is formulated as:

$$R(\tau) = R_{\text{acc}}(\tau) + R_{\text{format}}(\tau) + \lambda R_{\text{risk}}(\tau) + R_{\text{tool}}(\tau). \quad (8)$$

$R_{\text{acc}}(\tau)$ provides outcome-level supervision based on the final prediction $\hat{y}$, encouraging correct verification decisions. To reflect the asymmetric risk profile inherent in misinformation detection, we introduce an explicit risk-shaping term:

$$R_{\text{risk}}(\tau) = -\alpha\,\mathbb{I}_{\text{FP}}(\tau) - \gamma\,\mathbb{I}_{\text{FN}}(\tau), \quad (9)$$

where $\mathbb{I}_{\text{FP}}(\tau)$ and $\mathbb{I}_{\text{FN}}(\tau)$ indicate false-positive and false-negative outcomes, respectively. The coefficients $\alpha$ and $\gamma$ explicitly control the trade-off between precision and recall, allowing the detector's risk preference to be adjusted for different misinformation scenarios.

Beyond decision outcomes, reinforcement learning is constrained to valid and interpretable agent behaviors. Accordingly, $R_{\text{format}}(\tau)$ enforces well-formed outputs by requiring structured reasoning and final decisions to be enclosed within <think> and <answer> tags, thereby restricting policy optimization to well-defined agent trajectories.

In addition to supervising decision correctness and output validity, we introduce an action-level reward to explicitly regulate tool usage. Tool invocation is treated as a deliberate verification action and is rewarded only when it leads to a correct verification outcome, while unnecessary or ineffective tool use is penalized:

$$R_{\text{tool}}(\tau) = \begin{cases} r_{\text{tool}}^+, & \text{tool used and } R_{\text{acc}}(\tau) > 0, \\ -r_{\text{tool}}^-, & \text{tool used and } R_{\text{acc}}(\tau) \leq 0, \\ 0, & \text{otherwise.} \end{cases} \quad (10)$$

This design discourages indiscriminate tool usage while rewarding evidence-seeking behavior only when it contributes to correct verification. Consequently, GRPO optimizes verification-oriented agent behavior under uncertainty beyond raw label accuracy, including calibrated tool

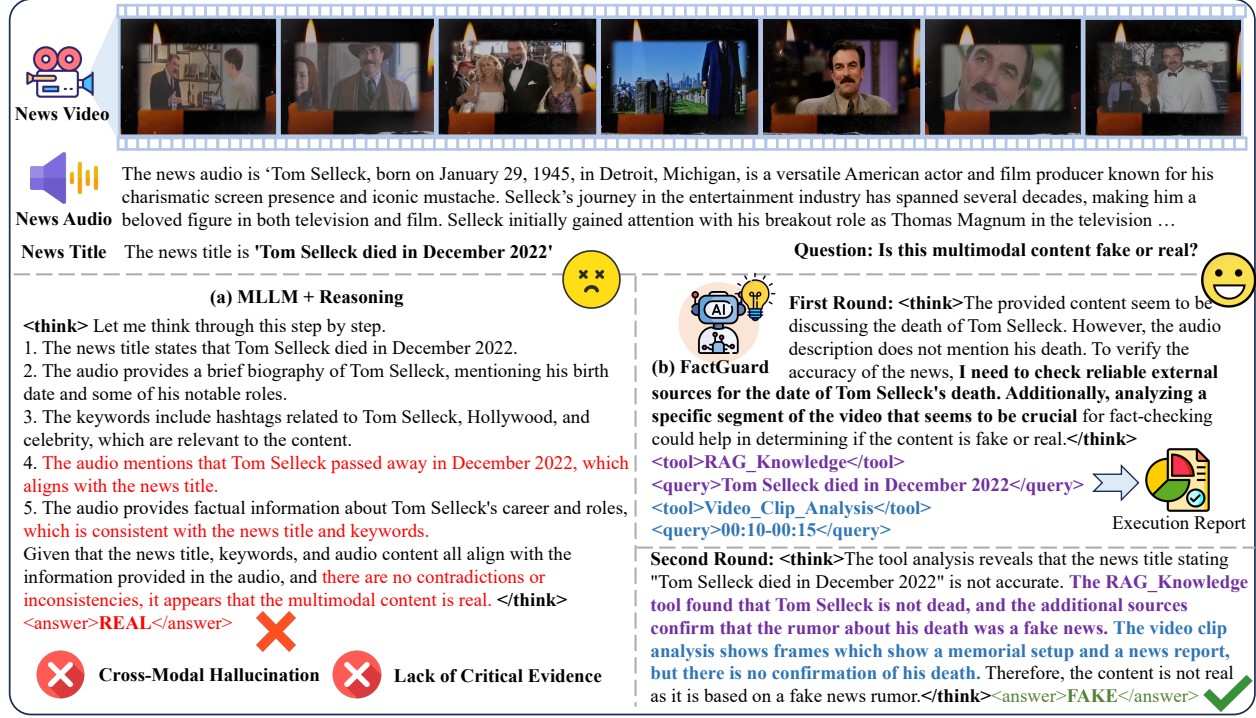

*Figure 3.* **Key advantages of FactGuard.** (a) MLLM-based methods with enhanced reasoning may induce cross-modal hallucination in ambiguous cases by over-relying on internally generated assumptions, treating them as grounded evidence without acquiring or validating critical supporting information. (b) FactGuard formulates misinformation verification as an uncertainty-aware, tool-assisted decision-making process that adaptively refines its conclusions, enabling reliable verification in open and dynamic environments.

invocation, evidence-grounded reasoning, and adaptive risk control. Unlike prior reinforcement learning methods that focus solely on output correctness, our approach treats misinformation detection as a decision-aware agentic problem, in which uncertainty handling and tool usage are first-class optimization objectives.

## 4. Experiment

### 4.1. Experimental Settings

**Baselines.** To comprehensively evaluate the performance of FactGuard, we compare it against three categories of baselines. **Discriminative Models** include the single-modality method, BERT (Koroteev, 2021) and multimodal methods such as TikTec (Shang et al., 2021), FANVM (Choi & Ko, 2021), SVFEND (Qi et al., 2023a), and FakingRec (Bu et al., 2024). **Zero-shot MLLMs** comprise closed-source models, including Gemini2-thinking (Gemini Team, 2023), Gemini-1.5-Pro (Team et al., 2024), GPT-4o (Achiam et al., 2023), GPT-o1-mini (Jaech et al., 2024a) and GPT-5 (Singh et al., 2025), as well as open-source models, including Qwen2.5-VL (Bai et al., 2025), InternVL2.5 (Chen et al., 2024), QVQ-72B (Qwen Team, 2024), InternVL2.5-MPO (Wang et al., 2024b), and DeepSeek-R1 (Guo et al., 2025), which are evaluated in a zero-shot setting. For models without native

video support, textual descriptions of news videos are used as substitutes. Finally, **Task-Aligned Reasoning Models** include Fact-R1 (Zhang et al., 2025a) and our method.

**Benchmarks.** We employ three widely adopted benchmark datasets: FakeSV (Qi et al., 2023a), FakeTT (Bu et al., 2024), and FakeVV (Zhang et al., 2025a). Following the protocol in (Qi et al., 2023a), we use a temporal split for testing, selecting the most recent 15% of samples from each dataset. In accordance with the Fact-R1 standard, all baseline models are trained on the training sets of FakeVV, FakeTT, and FakeSV. We report four standard evaluation metrics, including accuracy (ACC), precision, recall, and F1 score, to provide a comprehensive assessment of classification performance.

**Training Details.** Our model is trained on Qwen2.5-VL-7B using a two-stage pipeline consisting of supervised fine-tuning (SFT) followed by decision-aware reinforcement learning with GRPO. All experiments are conducted on 8 NVIDIA H100 GPUs. During GRPO training, each input prompt is unfolded into 8 candidate trajectories. The video-clip inspection tool can be invoked at most once per prompt, while external knowledge retrieval is unconstrained. We set the learning rate to $1 \times 10^{-6}$ and train the policy for one epoch. The maximum prompt length is set to 16,384

*Table 1.* Performance comparison on FakeSV, FakeTT and FakeVV datasets. We highlight the improvements achieved by FactGuard.

| Model | FakeSV | | | | FakeTT | | | | FakeVV | | | |
|---|---|---|---|---|---|---|---|---|---|---|---|---|
| | Acc | Prec | Rec | F1 | Acc | Prec | Rec | F1 | Acc | Prec | Rec | F1 |
| BERT (Koroteev, 2021) | 65.4 | 66.0 | 66.5 | 66.2 | 68.7 | 67.5 | 67.5 | 67.5 | 60.4 | 57.9 | 56.8 | 57.3 |
| TikTec (Shang et al., 2021) | 64.8 | 63.2 | 61.9 | 62.5 | 61.1 | 64.8 | 64.2 | 64.5 | 59.3 | 59.1 | 59.5 | 59.3 |
| FANVM (Choi & Ko, 2021) | 65.4 | 66.1 | 64.3 | 65.2 | 68.9 | 64.7 | 68.8 | 67.1 | 61.9 | 60.7 | 60.8 | 60.8 |
| SV-FEND (Qi et al., 2023a) | 67.1 | 67.4 | 66.3 | 66.8 | 67.6 | 72.2 | 69.0 | 70.6 | 70.9 | 71.4 | 71.3 | 71.3 |
| FakingRec (Bu et al., 2024) | 69.5 | 69.7 | 70.4 | 70.0 | 71.0 | 71.9 | 72.0 | 72.0 | 72.1 | 72.4 | 71.6 | 72.0 |
| Gemini2-thinking (Gemini Team, 2023) | 63.1 | 61.8 | 61.9 | 61.9 | 56.6 | 55.2 | 55.3 | 55.3 | 51.5 | 46.0 | 46.0 | 48.6 |
| Gemini-1.5-pro (Team et al., 2024) | 63.3 | 62.0 | 62.0 | 62.0 | 57.5 | 57.3 | 57.7 | 57.5 | 53.0 | 54.8 | 55.4 | 55.1 |
| GPT-4o (Achiam et al., 2023) | 66.6 | 65.2 | 64.7 | 64.9 | 57.9 | 57.8 | 62.9 | 60.2 | 56.0 | 60.4 | 35.0 | 44.3 |
| GPT-o1-mini (Jaech et al., 2024a) | 60.3 | 57.7 | 56.5 | 57.1 | 52.5 | 51.6 | 51.7 | 51.7 | 47.5 | 46.9 | 37.6 | 41.8 |
| GPT-5 (Singh et al., 2025) | 68.4 | 66.9 | 66.7 | 66.8 | 58.2 | 62.7 | 63.1 | 62.9 | 57.5 | 50.3 | 48.3 | 49.3 |
| DeepSeek-R1 (Guo et al., 2025) | 61.8 | 60.4 | 60.3 | 60.3 | 49.8 | 52.6 | 52.5 | 52.6 | 53.5 | 58.1 | 25.2 | 35.1 |
| Qwen2.5-VL-7B (Bai et al., 2025) | 55.6 | 55.5 | 55.7 | 55.6 | 54.9 | 54.0 | 54.1 | 54.0 | 52.9 | 51.1 | 51.1 | 51.1 |
| Qwen2.5-VL-72B (Bai et al., 2025) | 57.6 | 55.4 | 55.2 | 55.3 | 59.2 | 58.1 | 58.3 | 58.2 | 54.0 | 60.0 | 24.0 | 34.3 |
| QVQ-72B-preview (Qwen Team, 2024) | 60.8 | 59.0 | 58.8 | 58.9 | 58.1 | 54.0 | 52.8 | 53.4 | 53.5 | 52.6 | 52.6 | 52.6 |
| InternVL2.5-8B (Chen et al., 2024) | 49.8 | 52.6 | 52.5 | 52.6 | 43.9 | 44.0 | 44.0 | 44.0 | 53.5 | 58.5 | 24.0 | 34.0 |
| InternVL2.5-78B-MPO (Wang et al., 2024b) | 57.5 | 53.0 | 52.0 | 52.5 | 59.2 | 57.1 | 56.7 | 56.9 | 54.0 | 60.0 | 24.0 | 34.3 |
| Fact-R1 (Zhang et al., 2025a) | 75.6 | 77.7 | 72.0 | 74.7 | 74.4 | 77.8 | 68.3 | 72.7 | 81.2 | 84.5 | 76.4 | 80.3 |
| **FactGuard (Ours)** | **79.3** | **82.2** | **80.6** | **81.4** | **75.3** | 73.8 | **76.7** | **75.2** | **83.0** | **85.8** | **82.1** | **83.9** |

tokens, and the maximum response length is 768 tokens. The KL regularization coefficient is set to 0.04. Unless otherwise specified, we use a per-device batch size of 1 with no gradient accumulation. Additional implementation details are provided in Appendix A.1.

### 4.2. Main Results

**Comparison to State-of-the-Art Approaches.** As shown in Table 1, we evaluate FactGuard on the FakeSV, FakeTT, and FakeVV datasets and compare it with discriminative models, zero-shot MLLMs, and task-aligned reasoning models specifically trained or reinforced for misinformation detection. All results are reported using accuracy, precision, recall, and F1-score. Overall, FactGuard consistently outperforms existing methods across all datasets in terms of both accuracy and F1 score.

For discriminative approaches, FakingRec yields the strongest performance, benefiting from its effective modeling of fine-grained multimodal correlations and its exploitation of user–content interaction patterns, which are particularly suited for recommendation-style verification. For zero-shot MLLMs, GPT-5 achieves the strongest overall performance, particularly on FakeSV and FakeTT, suggesting stronger multimodal reasoning and verification ability. However, this advantage is not consistent across all metrics: on FakeVV, GPT-5 obtains the highest accuracy, while Gemini-1.5-pro achieves a higher F1 score. This indicates that stronger general-purpose models may still suffer from dataset-specific biases and imbalanced misinformation detection. Within the same model family, more advanced models generally perform better, although their effectiveness

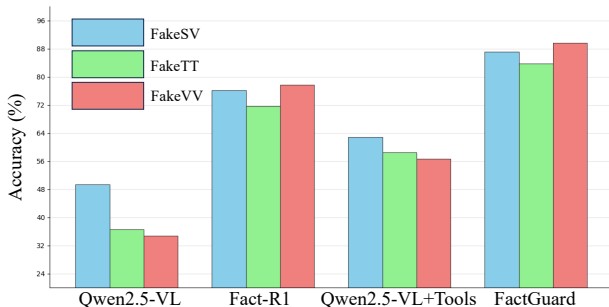

*Figure 4.* **Analysis of model reasoning.** Representative reasoning traces under correct predictions show that FactGuard produces more coherent and evidence-grounded reasoning than Qwen2.5-VL and Fact-R1, highlighting improved interpretability.

varies across datasets due to differences in task difficulty and evidence availability.

Compared to discriminative methods, task-aligned reasoning approaches, including Fact-R1 and our method, not only provide improved interpretability through explicit reasoning processes but also achieve substantially higher verification accuracy. While Fact-R1 significantly improves performance via reinforcement learning, FactGuard consistently delivers superior results. As illustrated in Figure 3(b), this advantage stems from FactGuard's ability to avoid cross-modal hallucination in ambiguous cases by explicitly recognizing uncertainty and adaptively acquiring supporting evidence through tool-assisted, iterative decision making.

**Explainability Analysis.** Compared to discriminative models, multimodal large language model–based ap-

*Table 2.* Stage-wise inference latency breakdown. The latency is reported as the average time per sample in seconds.

| Method | Stage-1 | Stage-2 | FactProbe | ClipScout | Total ↓ |
|---|---|---|---|---|---|
| Fact-R1 | 4.4 | – | – | – | 4.4 |
| FactGuard | 3.2 | 3.1 | 4.7 | 0.3 | 11.3 |

*Table 3.* Cost-sensitive precision and recall achieved by FactGuard on FakeSV and FakeVV under different asymmetric error cost ratios.

| Cost Ratio | FakeSV | | FakeVV | |
|---|---|---|---|---|
| $(\alpha : \gamma)$ | Precision | Recall | Precision | Recall |
| 1:2 | 80.8 | **82.1** | 83.2 | **83.7** |
| 1:1 | 82.2 | 80.6 | 85.8 | 82.1 |
| 2:1 | **83.2** | 75.0 | **86.6** | 79.3 |

proaches not only produce final predictions but also generate explicit reasoning traces, substantially improving interpretability. Accordingly, we adopt **GPT-4o** as an automatic evaluator to assess the quality of model-generated reasoning and to provide fine-grained reasoning accuracy scores. The evaluation considers multiple complementary dimensions, including faithfulness to the provided multimodal evidence, logical consistency of the reasoning chain, and the ability to capture salient misinformation patterns.

To ensure a fair and meaningful evaluation, we restrict the analysis to instances where the predicted labels are correct, thereby preventing incorrect predictions from confounding the assessment of reasoning quality. As illustrated in Figure 4, we compare Qwen2.5-VL and Fact-R1 without tool assistance against their tool-augmented counterparts, Qwen2.5-VL with tools and FactGuard. The results indicate that both reinforcement learning and external tool integration contribute to improved reasoning capability over the base model. Reinforcement learning enhances the model's ability to generate structured and consistent reasoning, while tool integration further strengthens evidence grounding. Notably, their combination yields a complementary effect: FactGuard consistently exhibits the strongest reasoning performance, producing coherent, evidence-grounded, and logically sound inference chains across diverse cases.

**Cost-Sensitive Risk Analysis.** We report the precision and recall on the FakeSV and FakeVV datasets to analyze the effect of cost-sensitive risk modeling. As shown in Table 3, the cost ratio $\alpha : \gamma$ plays a critical role in shaping the trade-off between precision and recall. When $\alpha$ is increased, the model places a higher penalty on false positives, leading to improved *Fake* precision at the cost of a modest reduction in recall. Conversely, emphasizing $\gamma$ encourages the model to reduce false negatives, resulting in higher recall but lower precision. This behavior highlights the flexibility of the proposed cost-sensitive risk function in regulating decision

*Table 4.* Ablation study of FactGuard on FakeSV and FakeTT datasets. We report Accuracy (%) and F1-score (%) to evaluate the contribution of each component.

| Model | FakeSV | | FakeTT | |
|---|---|---|---|---|
| | Acc | F1 | Acc | F1 |
| **FactGuard (Ours)** | **79.3** | **81.4** | **75.3** | **75.2** |
| w/o SFT | 73.1 | 74.7 | 71.6 | 68.2 |
| w/o RL | 62.0 | 63.2 | 67.9 | 65.7 |
| w/o $R_{tool}$ | 77.7 | 78.5 | 74.7 | 72.9 |
| w/o $R_{risk}$ | 78.5 | 78.9 | 74.9 | 73.8 |
| *Base (Qwen2.5-VL-7B+tool)* | 57.6 | 60.6 | 56.8 | 55.4 |

preferences under asymmetric error costs. Unless otherwise specified, we adopt a balanced cost ratio of $\alpha : \gamma = 1 : 1$ in all other experiments.

Unlike conventional misinformation detectors that optimize a fixed objective, FactGuard enables explicit control over the precision–recall balance, allowing the verification policy to be adapted to different deployment requirements. In practice, this is particularly important for misinformation detection, where the relative costs of false alarms and missed misinformation can vary substantially across application scenarios. For example, high-precision settings are desirable in content moderation to avoid unjustified censorship, whereas high-recall configurations are preferable in early-warning or monitoring systems to minimize the spread of harmful content. These results demonstrate that FactGuard offers a principled and practical approach to risk-aware verification, effectively bridging model optimization with real-world decision-making constraints.

**Efficiency and Latency Analysis** To better understand the inference efficiency of FactGuard, we provide a stage-wise latency breakdown and compare it with the single-pass baseline Fact-R1. Table 2 reports the average inference latency per sample, decomposed into Stage-1 reasoning, Stage-2 reasoning, and external tool execution.

As shown in Table 2, FactGuard introduces additional latency compared with the single-pass baseline, mainly due to the second-stage reasoning and external tool calls. Among the tool components, FactProbe contributes the dominant overhead, while ClipScout incurs only a lightweight cost. This is because FactProbe currently relies on API-based web retrieval, where the latency is largely affected by network conditions rather than the model inference itself.

Importantly, this overhead is not intrinsic to the proposed framework. In practical deployments, the API-based retrieval module can be replaced with a locally hosted knowledge base. Under this setting, the tool-related overhead becomes minimal, and the additional cost of FactGuard is primarily from the second-stage reasoning, leading to a moderate latency increase compared with Fact-R1.

*Table 5.* Performance of GRPO variants on FakeSV and FakeTT.

| Variant | FakeSV | | FakeTT | |
|---|---|---|---|---|
| | Acc | F1 | Acc | F1 |
| *(a)* w/. Rollouts = 6 | 77.3 | 78.9 | 72.6 | 69.4 |
| *(b)* w/. Rollouts = 8 (Ours) | 79.3 | 81.4 | **75.3** | **75.2** |
| *(c)* w/. Rollouts = 12 | **79.5** | **81.5** | 75.2 | 74.3 |
| *(d)* w/. $\beta$ = 0.02 | 78.2 | 79.5 | 74.6 | 73.4 |
| *(e)* w/. $\beta$ = 0.04 (Ours) | **79.3** | **81.4** | **75.3** | **75.2** |
| *(f)* w/. $\beta$ = 0.06 | 78.8 | 80.9 | 74.8 | 73.7 |

*Table 6.* Analysis of tool-use behavior. We report the average number of FactProbe calls per sample under different training settings.

| Model | Avg. FactProbe Calls | Tool-use Behavior |
|---|---|---|
| Base (Qwen2.5-VL-7B) | 4–5 | Redundant / Unstable |
| + GRPO | 3–4 | Partially Improved |
| + Decision-aware RL | 1–2 | Efficient / Stable |

Based on this balance, we adopt 8 rollouts and $\beta$ = 0.04.

We argue that this efficiency–performance trade-off is necessary for open-world, knowledge-intensive misinformation verification, where a single-pass prediction is often insufficient. By selectively invoking external tools only when additional evidence is needed, FactGuard achieves a favorable balance between verification accuracy, adaptive evidence acquisition, and inference efficiency.

### 4.3. Ablation Studies

**Variants of component ablations.** Table 4 presents the ablation results of FactGuard on the FakeSV and FakeTT datasets. The full model consistently achieves the best performance, indicating that all components contribute positively to misinformation detection. Removing either supervised fine-tuning (SFT) or reinforcement learning (RL) leads to substantial performance degradation, with the absence of RL causing the most severe drop, highlighting its critical role in optimizing multi-step verification and decision-making. In contrast, ablating the tool-related reward ($R_{\text{tool}}$) or the risk-aware reward ($R_{\text{risk}}$) results in moderate but consistent performance declines, suggesting that these rewards provide complementary benefits by improving evidence grounding and uncertainty calibration.

We find that reinforcement learning without prior SFT yields inferior performance, consistent with Video-Star (Yuan et al., 2025). Direct RL causes the policy to collapse toward tool-free reasoning, as it must simultaneously explore complex reasoning trajectories and tool-use decisions under distributional mismatch. This motivates our two-stage training scheme, where SFT establishes structured evidence-seeking behaviors before RL refinement.

**Variants of different GRPO hyperparameters.** As shown in Table 5, increasing the number of rollouts generally improves performance by yielding more reliable advantage estimates, with diminishing returns beyond 8 rollouts, while further increasing the number of rollouts incurs additional computational overhead with limited efficiency gains. We also observe that the KL coefficient $\beta$ exhibits a clear trade-off: overly small values lead to insufficient regularization, while larger values overly constrain policy updates.

**Ablation of Tool-aware Reward.** Although the tool-aware reward brings only moderate gains in overall accuracy, we find that its primary role is to regulate tool-use behavior during reinforcement learning. To better understand this effect, we analyze the average number of FactProbe invocations per sample under different training settings. As shown in Table 6, the base Qwen2.5-VL-7B model tends to overuse external tools, often producing redundant and unstable tool calls. Standard GRPO partially alleviates this issue, but still results in unnecessary tool invocations. In contrast, our decision-aware RL reduces the average number of FactProbe calls to only 1–2 essential invocations in most cases, leading to a more efficient and stable inference policy.

These results indicate that the tool reward discourages undesirable behaviors such as redundant tool calls and reward hacking. Instead of encouraging the model to invoke tools indiscriminately, the proposed reward design guides the model to acquire external evidence only when necessary.

## 5. Conclusion

In this work, we presented **FactGuard**, an agentic framework for video misinformation detection that formulates verification as an uncertainty-aware, iterative decision-making process. By integrating multimodal reasoning with selective, tool-assisted evidence acquisition, FactGuard addressed key limitations of existing approaches, including over-reliance on internally generated assumptions and the failure to obtain critical supporting evidence in ambiguous cases. We further introduced a two-stage training strategy that combines agentic Chain-of-Thought supervised fine-tuning with decision-aware reinforcement learning, enabling high-quality reasoning, calibrated tool usage, and explicit control over asymmetric verification risks. Extensive experiments on three public benchmarks demonstrate that FactGuard consistently outperforms strong discriminative baselines and MLLM-based reasoning models, including reinforcement learning–enhanced methods, in both prediction accuracy and reasoning quality.

## Acknowledgements

The authors would like to thank the anonymous reviewers for their insightful comments and constructive suggestions,

which helped improve the quality of this paper.

This work was supported in part by the National Natural Science Foundation of China under Grant No. 62406310, and in part by the Innovation Research Program of ICT under Grant No. E261070.

## Impact Statement

This work aims to advance research in multimodal reasoning, agentic decision-making, and video misinformation detection. The proposed framework is developed solely for research purposes and is intended to support the study of reliable and interpretable verification systems. All datasets used in this work are publicly available and are employed in accordance with their respective licenses and intended usage scopes. Any additional data constructed for training or evaluation are synthetically generated or obtained through automated processes, and are used exclusively for research purposes. While the techniques studied in this work could potentially be applied in real-world verification or content moderation scenarios, we emphasize that our contributions are intended as research tools rather than deployment-ready systems. We do not foresee significant negative societal impact when the proposed methods are used responsibly and within their intended research scope.

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

# Appendix

In this appendix, we provide more experimental details, related work, tool library, and discussions for a comprehensive evaluation and understanding of our method. Detailed contents are as follows:

## A. Experiment

### A.1. More Experimental Details

**Datasets.**   A variety of datasets have been developed to facilitate research on video-based misinformation detection. Early datasets typically focused on narrow domains, such as medical misinformation (Hou et al., 2019) or COVID-19-related content (Knuutila et al., 2021), and often covered multiple languages. While valuable for domain-specific analysis, these datasets are generally limited in scale, topical diversity, and long-term public availability.

More recent benchmarks have shifted toward large-scale, short-video misinformation detection. In particular, FakeSV (Qi et al., 2023a) consists of short news-style videos collected from social media platforms, paired with textual metadata and crowd-sourced annotations. FakeTT (Bu et al., 2024) focuses on TikTok videos and incorporates multimodal signals including visual content, audio transcripts, and user interaction features. FakeVV (Zhang et al., 2025a) focuses on more complex and ambiguous misinformation scenarios and offers richer multimodal signals and annotations, enabling more comprehensive evaluation of reasoning-oriented verification models.

Together, these datasets provide complementary coverage in terms of content sources, multimodal structure, and verification difficulty, enabling comprehensive evaluation of video misinformation detection systems under diverse real-world conditions. Accordingly, all experimental evaluations in this work are conducted on FakeSV, FakeTT, and FakeVV.

**Training Details.**   To support agentic reasoning and tool-aware decision making, we first construct a multimodal agentic Chain-of-Thought (CoT) dataset tailored for video misinformation detection. Starting from the training sets of FakeSV, FakeTT, and FakeVV, we employ a stronger teacher model, Qwen2.5-VL-72B, to generate agentic reasoning trajectories under carefully designed prompts. These prompts explicitly encourage uncertainty assessment, tool-selection intent, evidence grounding, and final verification decisions conditioned on multimodal inputs.

Given the multimodal input and the ground-truth label, the teacher model produces full agent trajectories, including intermediate reasoning steps, tool invocation decisions, and evidence-aware conclusions. To ensure data quality, we apply a two-stage filtering process. First, rule-based validation removes malformed trajectories, such as missing reasoning structure, invalid tool actions, or incorrect final decisions. Second, a subset of samples is manually inspected to further eliminate low-quality or hallucinated reasoning traces. Only high-quality, coherent, and evidence-grounded trajectories are retained for supervised fine-tuning.

We perform agentic supervised fine-tuning (SFT) on Qwen2.5-VL-7B-Instruct using the curated agentic CoT dataset. The objective of this stage is to inject structured reasoning patterns, explicit uncertainty awareness, and tool-invocation behaviors into the base model prior to reinforcement learning.

Training is conducted with mixed-precision (BF16) and DeepSpeed optimization. We use a per-device batch size of 1 with gradient accumulation and train the model for one epoch. Gradient checkpointing and FlashAttention are enabled to reduce memory consumption. This SFT stage provides essential inductive bias for evidence-seeking and multi-step reasoning, which we find to be critical for stabilizing and guiding subsequent reinforcement learning.

Following supervised fine-tuning, we further optimize FactGuard using decision-aware reinforcement learning with Group Relative Policy Optimization (GRPO). All reinforcement learning experiments are conducted on 8 NVIDIA H100 GPUs.

During GRPO training, each input prompt is unfolded into 8 candidate trajectories, capturing diverse reasoning paths, tool-invocation behaviors, and final verification outcomes. The video-clip inspection tool is restricted to at most one invocation per prompt, encouraging selective and deliberate visual evidence acquisition, while external knowledge retrieval remains unconstrained. Unless otherwise specified, the model is configured with a balanced cost setting ($\alpha : \gamma = 1 : 1$), encouraging neutral treatment of false positives and false negatives during trajectory generation.

We set the learning rate to $1 \times 10^{-6}$ and train the policy for one epoch. The maximum prompt length is set to 16,384 tokens and the maximum response length is 768 tokens. The KL regularization coefficient is fixed to 0.04 to stabilize policy updates against a frozen reference model. We use a per-device batch size of 1 without gradient accumulation.

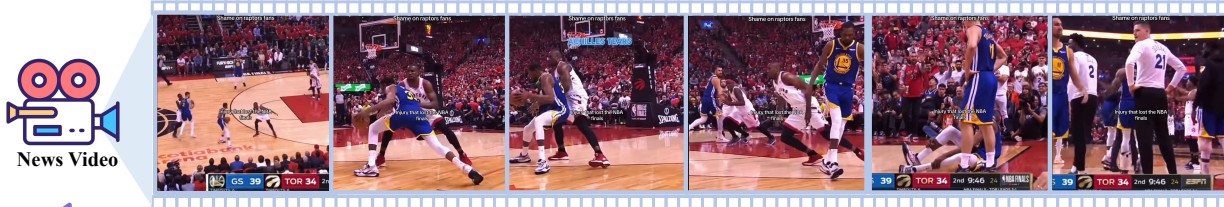

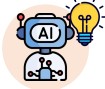

**News Video**

**News Audio**

The news audio is 'his leg Ibaka goes up and it's down down the other end and Durant grabbing that right leg it's the right calf that put him out his teammates going over to check on him cheering I'm surprised by that and and the players are telling him not to be doing that no question that Kyle Lowry Pascal Siak for Serge Ibaka telling the crowd no we are not going to cheer when this…

**News Title**     The news title is **'Toronto Bar Cheer at Kevin Durant's Injury'**          **Question: Is this multimodal content fake or real?**

**FactGuard**

**First Round: \<think\>**To determine if the provided content is fake or real, I need to verify the factual claims made in the news title, keywords, and audio. The most critical claim is that the Raptors fans cheered when Kevin Durant injured his Achilles tendon during the NBA Finals. This claim can be fact-checked using reliable external sources. Additionally, analyzing a specific segment of the video that shows the injury and the fans' reaction could provide visual evidence to support or refute the claim. Therefore, I will use the RAG_Knowledge tool to verify the factual claim and the Video_Clip_Analysis tool to analyze the relevant segment of the video. **\</think\> \<tool\>RAG_Knowledge\</tool\>\<query\>Did the Toronto Raptors fans cheer when Kevin Durant injured his Achilles tendon during the NBA Finals?\</query\>\<tool\>Video_Clip_Analysis\</tool\>\<query\>00:10-00:20\</query\>"**

**Second Round: \<think\>**To determine if the content is real, I will analyze the information from both the first-turn input and the tool outputs.
**1. RAG_Knowledge Tool Output:** - The top source from the RAG_Knowledge tool confirms that Toronto Raptors fans did indeed cheer when Kevin Durant injured his Achilles tendon during the NBA Finals. This aligns with the news title and keywords provided. - Additional sources further support this claim, with Kevin Durant himself addressing the controversy and the incident being widely reported.
**2. Video_Clip_Analysis Tool Output:** - The video clip analysis shows a sequence of frames that depict the moment of Durant's injury and the subsequent reaction of the crowd. The frames show Durant falling to the ground, and the crowd's reaction, which appears to be cheering.
**3. Consistency and Credibility:** - The information from the RAG_Knowledge tool and the video clip analysis is consistent, with multiple sources confirming the event and the fans' Given the consistency between the information from the RAG_Knowledge tool, the video clip analysis, and the visual evidence in the video frames, the content appears to be real.**\</think\> \<answer\>REAL\</answer\>"**

*Figure 5.* Additional Case Study of FactGuard.

*Table 7.* Generalization performance on FakeVV under cross-dataset and in-domain settings.

| Method | Train Dataset | Test Dataset | Acc. ↑ |
|---|---|---|---|
| FactR1 | FakeSV + FakeTT | FakeVV | 65.4 |
| FactGuard | FakeSV + FakeTT | FakeVV | 72.7 |
| FactR1 | FakeSV + FakeTT + FakeVV | FakeVV | 81.2 |
| FactGuard | FakeSV + FakeTT + FakeVV | FakeVV | 83.0 |

This reinforcement learning stage refines verification decisions under task-aligned rewards, explicitly calibrating tool usage and asymmetric error costs, and further strengthens evidence-grounded reasoning beyond supervised imitation.

**A.2. More Results**

As illustrated in Figure 5, when confronted with uncertain or insufficient evidence, FactGuard adaptively invokes multiple tools to refine its verification process. Specifically, FactGuard first calls *FactProbe* to retrieve external knowledge and verify the factual validity of the claimed event. It then employs *ClipScout* to perform focused visual analysis, attending to the reactions of surrounding spectators in the video. By integrating the retrieved factual evidence with the localized visual cues, FactGuard is able to conduct a more informed and reliable reasoning process, ultimately leading to a more accurate and well-supported conclusion.

**Generalization to Unseen Misinformation Categories**     To evaluate whether FactGuard can generalize beyond the training distribution, we conduct both cross-dataset and in-domain experiments on FakeVV. Specifically, we consider two training settings: (i) a cross-domain setting, where models are trained on FakeSV and FakeTT and evaluated on FakeVV, and (ii) an in-domain setting, where models are trained on FakeSV, FakeTT, and FakeVV and evaluated on FakeVV.

As shown in Table 7, FactGuard substantially outperforms FactR1 in the cross-dataset setting, improving accuracy from

*Table 8.* Impact of teacher model choice on student performance on FakeSV.

| Teacher Model | Training Stage | Acc. ↑ |
|---|---|---|
| Qwen2.5-VL-72B | SFT only | 62.0 |
| InternVL2.5-78B-MPO | SFT only | 60.8 |
| Qwen2.5-VL-72B | SFT + RL | 79.3 |
| InternVL2.5-78B-MPO | SFT + RL | 78.7 |

*Table 9.* Ablation study on the contribution of external tools and RL on FakeSV.

| Method | FactProb | ClipScout | RL | Acc. ↑ |
|---|---|---|---|---|
| Qwen2.5-VL-7B (base) | × | × | × | 55.6 |
| + FactProb | ✓ | × | × | 56.7 |
| + ClipScout | × | ✓ | × | 55.9 |
| + FactProb + ClipScout | ✓ | ✓ | × | 57.6 |
| FactGuard w/o FactProb | × | ✓ | ✓ | 76.1 |
| FactGuard w/o ClipScout | ✓ | × | ✓ | 78.2 |
| FactGuard (full) | ✓ | ✓ | ✓ | 79.3 |

65.4% to 72.7%. This indicates that the proposed agentic verification framework generalizes better to unseen misinformation categories, where target-domain training data are not available. When FakeVV is included during training, both models achieve higher accuracy, while the performance gap between FactGuard and FactR1 becomes smaller, i.e., 83.0% versus 81.2%. This suggests that FactR1 benefits more from direct exposure to target-domain samples, whereas FactGuard shows stronger robustness under distribution shifts by acquiring and integrating external evidence during verification.

**Impact of Teacher Model Choice.** The SFT stage initializes the student model with structured reasoning and tool-use patterns distilled from a stronger teacher model. To examine whether the final performance depends on a specific teacher, we further generate SFT data using InternVL2.5-78B-MPO and compare it with our default teacher, Qwen2.5-VL-72B, on the FakeSV dataset. The results are shown in Table 8.

As shown in Table 8, the SFT-only performance varies with the choice of teacher model, indicating that supervised distillation is affected by the quality and style of the teacher-generated trajectories. However, after applying decision-aware reinforcement learning, the performance gap between different teachers becomes much smaller, with Qwen2.5-VL-72B and InternVL2.5-78B-MPO achieving 79.3% and 78.7% accuracy, respectively. This suggests that the final performance of FactGuard is not tightly bound to a specific teacher model. Instead, the task-specific reward optimization in the RL stage plays a central role in refining the student model's verification policy beyond imitation of the teacher trajectories.

**Ablation of External Tools and RL.** To clarify the roles of different tools and their interaction with reinforcement learning, we conduct a comprehensive ablation study on FakeSV. As shown in Table 9, we compare the base Qwen2.5-VL-7B model with different combinations of FactProb, ClipScout, and RL.

The results reveal three observations. First, both FactProb and ClipScout individually improve the base model, and combining them brings further gains, indicating that the two tools provide complementary evidence. Second, removing either tool from FactGuard leads to a clear performance drop, showing that both tools contribute to the final verification performance. In particular, removing FactProb causes a larger drop than removing ClipScout, suggesting that external factual evidence is especially important for knowledge-intensive misinformation verification. Third, all RL-based variants substantially outperform their non-RL counterparts, demonstrating that the improvement does not come from tool access alone. Instead, decision-aware RL enables the model to better decide when and how to use external evidence for final verification.

Overall, these results show that FactGuard's gains stem from the integrated design of external tool augmentation and learned tool-use policies, rather than from simply adding tools or applying RL in isolation.

## B. Tool Library Details

FactGuard is equipped with a lightweight yet effective tool library to support evidence acquisition during agentic verification. The tool library is designed to complement the model's internal multimodal reasoning by providing access to external factual knowledge and localized visual evidence. In this work, we implement two core tools: an external knowledge retrieval tool (**FactProbe**) and a video clip inspection tool (**ClipScout**). These tools are selectively invoked by the agent when internal reasoning alone is insufficient for confident verification.

### B.1. External Knowledge Retrieval (FactProbe)

FactProbe implements a retrieval-augmented generation (RAG) pipeline for external factual verification. Given a structured factual query produced by the agent during the first-stage reasoning, the tool issues a web search request via a commercial search API (Serper) to retrieve a small set of relevant results. To reduce noise, only organic search results are retained, and user-generated or social media sources are excluded.

For each query, the tool collects the top-ranked results and extracts their titles, snippets, and source links. The retrieved content is then aggregated into a compact textual report that summarizes the external evidence relevant to the queried claim. Rather than performing full document reading, FactProbe adopts a lightweight retrieval-and-snippet strategy to balance evidence coverage and efficiency. The resulting report is passed to the second-stage reasoning process as auxiliary textual input, enabling the model to incorporate externally grounded information during verification.

### B.2. Video Clip Temporal Inspection (ClipScout)

ClipScout provides targeted temporal inspection of video content by extracting representative frames from a specified time interval. When the agent invokes ClipScout, it outputs a precise temporal query (e.g., a start–end timestamp) corresponding to the most evidence-bearing segment of the video.

The tool decodes the video stream and uniformly samples a small number of frames (four by default) within the queried interval. These frames are then arranged into a $2 \times 2$ visual grid to form a compact visual summary. To control computational and token overhead, the grid is resized such that its maximum spatial dimension does not exceed a fixed resolution threshold.

The resulting composite image is appended to the multimodal input during the refinement stage, allowing the model to reason over localized visual evidence without processing the entire video. ClipScout is constrained to at most one invocation per instance, ensuring that temporal inspection remains selective and aligned with deliberate evidence acquisition rather than exhaustive visual scanning.

## C. Prompt Templates

We present the prompt templates used in our framework, including the two-stage prompting strategy for FactGuard and the auditing prompt employed by GPT-4o. FactGuard operates in a structured two-turn manner: the first turn determines whether external evidence is required and selects appropriate tools, while the second turn produces the final verification result conditioned on the gathered evidence. In addition, GPT-4o is used as an external auditor to assess the evidence grounding of the generated reasoning.

**FactGuard's Two-Turn Prompting.** As shown in Figure 6, FactGuard adopts a two-turn prompting strategy to enable explicit evidence-seeking and grounded verification. In the first turn, the model analyzes the input claim and decides whether external tools should be invoked to acquire additional evidence. In the second turn, FactGuard integrates the retrieved evidence with its internal reasoning to produce a final prediction and explanation. This design explicitly separates evidence acquisition from answer generation, reducing reliance on unsupported internal assumptions.

**GPT-4o's Reasoning Audit Prompt.** To assess the faithfulness, logical consistency, and evidence grounding of model-generated reasoning, we adopt GPT-4o as an external auditor with a dedicated evaluation prompt. Given the model's prediction, reasoning trace, and associated evidence, GPT-4o evaluates whether the conclusion is logically coherent, internally consistent, and supported by explicit evidence rather than speculative or internally generated assumptions, as illustrated in Figure 7. This auditing procedure provides an additional layer of quality control to evaluate the quality and interpretability of reasoning.

TOOL_SELECTION_TEMPLATE = (
    "{Question}\n"
    "You are a professional disinformation detection agent. Please carefully analyze the provided media content and the question.\n"
    "You can use the following tools:\n\n"
    "1. [RAG_Knowledge]: When you need external knowledge, use this tool to verify specific factual claims, dates, or events "
    "against reliable external sources (web search + webpage reading). The <query> content must directly express the factual "
    "claim you want to check, rather than a meta description about what the video or post says.\n"
    "2. [Video_Clip_Analysis]: When certain segments of the video seem especially important, use this tool to focus on and analyze a
specific time range. Important: The query for this tool MUST be a precise time interval (format like 00:05-00:10).\n\n"
    "**Instructions**:\n"
    "- First, analyze the input and decide whether any tools are needed, and which ones can best assist your judgment.\n"
    "- You may select one or multiple tools, and you may call the same tool more than once if it helps your analysis.\n"
    "- If you can confidently make a judgment without using any tools, output exactly: <tool>None</tool><query></query>.\n"
    "- Tool names MUST be exactly one of: RAG_Knowledge, Video_Clip_Analysis, or None.\n\n"
    "- **Video_Clip_Analysis can be called at most ONCE, and ONLY for the single most important segment of the video that is
essential for fact-checking.**\n\n"
    "Please strictly follow the XML-style format below for your decision (do not output example content; fill in your actual reasoning
and queries):\n"
    "<think>\n"
    "...Explain here whether tools are needed and why, and how you plan to use them...\n"
    "</think>\n"
    "<tool>The name of the tool you select</tool><query>The specific query content for this tool</query>\n"
    "(If multiple tool calls are needed, continue outputting additional <tool>...</tool><query>...</query> pairs on new lines.)"
)

ANSWER_TEMPLATE = (
    "Based on BOTH the complete first-turn input (original media and user question) "
    "AND the **Tool Analysis Reports** above, determine if the content is FAKE or REAL.\n"
    "You must consider information from both the first-turn input and the tool outputs.\n"
    "Think step by step inside <think>...</think> tags.\n"
    "Finally, output exactly one label inside <answer>...</answer> tags: FAKE or REAL."
)

*Figure 6.* **FactGuard Two-Turn Prompting.**

```
"""
Core evaluation logic: assessing the factual reliability, logical coherence, and utilization of multimodal evidence in the reasoning
process
"""
prompt = f"""
You are an expert in multimodal content safety and reasoning auditing. Your task is to assess the reliability of the AI model's
reasoning process.

[Case Information]
Question: {case.get('problem')}
Audio/Textual Evidence: {case.get('news_audio', 'No audio content provided')}
Model Prediction: {case.get('prediction')} (known to be correct)

[Model Reasoning]
{case.get('agent_process')}

[Evaluation Dimensions (0.0 – 1.0)]
1. Faithfulness: Does the reasoning remain faithful to the provided inputs? Does it hallucinate information not present in the audio or
text?
2. Consistency: Are the reasoning steps logically coherent and well-connected? Are there any logical leaps?
3. Insight: Does the model identify common patterns or cues associated with misinformation?

Please return the result in JSON format:
{
    "score": 0.00,
    "reason": "A brief explanation of the strengths and weaknesses of the reasoning"
}
"""
```

*Figure 7.* **GPT-4o reasoning audit prompt.** The prompt evaluates whether a model's reasoning and prediction are supported by explicit and relevant evidence, serving as an external assessment of reasoning faithfulness.

