# OpenReview forum: "FactGuard: Agentic Video Misinformation Detection via Reinforcement Learning"
_ICML.cc/2026/Conference — ICML 2026 regular_

### Official Review · Reviewer_L4aL · 2026-03-09

**Soundness:** 2
**Presentation:** 2
**Significance:** 3
**Originality:** 2
**Overall Recommendation:** 3
**Confidence:** 4

**Summary:**

This paper targets the task of video misinformation detection and proposes the FactGuard framework. FactGuard formulates video misinformation detection as an interactive decision-making process of an agent, which can identify task ambiguity and proactively invoke external tools to obtain evidence, thereby conducting iterative reasoning to determine the authenticity of videos. The method achieves competitive performance on several benchmark datasets.

**Compliance With Llm Reviewing Policy:**

Affirmed.

**Final Justification:**

Thank you for the authors’ rebuttal. Upon further consideration, I also feel that the proposed FactGuard framework is mainly an application of the standard agent paradigm to the video misinformation detection task, and the novelty is limited. Therefore, I will maintain my score.

**Key Questions For Authors:**

Please refer to the weaknesses.

**Limitations:**

Yes, the limitations have been discussed in this paper.

**Strengths And Weaknesses:**

Strengths:

1. FactGuard formulates video misinformation detection as an uncertainty-aware iterative decision-making process, enabling the model to proactively call external tools for evidence when information is insufficient.
2. The method achieves superior experimental results on three benchmark datasets: FakeSV, FakeTT, and FakeVV.

Weaknesses:

1. As an iterative agent framework, FactGuard may incur relatively high inference latency and computational cost. The paper lacks a systematic comparison with existing methods in terms of inference efficiency, and does not provide statistics such as the actual tool-calling frequency.
2. The ClipScout tool samples only 4 frames by default. Such a low sampling rate may cause key tampering evidence to be missed. It is recommended to further analyze or discuss its impact.
3. The paper does not compare FactGuard with the latest closed-source models equipped with search capabilities.
4. The value ranges of the four reward terms in Equation (8) are not specified, and the details on how the SFT dataset is constructed are not sufficiently clear.
5. In Table 3, on the FakeSV dataset, removing SFT (w/o SFT) leads to a 6.2% performance drop, while removing RL (w/o RL) causes a 17.3% drop. The second paragraph of the Ablation Studies section states that “reinforcement learning without prior SFT leads to reasoning collapse,” but it is unclear why the performance degradation of w/o RL is so large.

---

> ### Author Rebuttal · Authors · 2026-03-31
>
> We thank the reviewer for recognizing our formulation of video misinformation detection as an uncertainty-aware, iterative decision-making process, and its effectiveness in leveraging external tools for evidence-based reasoning.
>
> ## Q1: Stage-wise Inference Efficiency and Tool Usage Analysis
>
> We thank the reviewer for the suggestion to provide a more detailed efficiency analysis. We present a stage-wise breakdown of the inference time for FactGuard.
>
> Table 1 shows the average latency (s) per sample, decomposed into different stages.
>
> | Method     | Stage-1 Reasoning | Stage-2 Reasoning | FactProb  | ClipScout | Total ↓ (s) |
> |------------|------------------|-------------------|-----------|-----------|-------------|
> | FactR1     | *4.4*            | –                 | –         | –         | *4.4*       |
> | FactGuard  | *3.2*            | *3.1*             | *4.7*     | *0.3*     | *11.3*      |
>
> As shown in Table 1, the additional latency of FactGuard mainly comes from second-stage reasoning and external tool calls, with FactProb being the dominant overhead. This overhead is not intrinsic, as it largely stems from API-based retrieval and can be mitigated with a local knowledge base. Compared to single-pass models such as FactR1, FactGuard introduces moderate cost while achieving substantially better performance, especially for knowledge-intensive cases.
>
> We further analyze tool usage frequency. ClipScout is typically invoked at most once per sample. In contrast, the base model (Qwen2.5-VL-7B) often overuses FactProb with 4–5 redundant calls, while our decision-aware RL reduces this to 1–2 essential calls in most cases. This leads to more efficient and stable inference.
>
> ---
>
> ## (2) ClipScout Frame Sampling Strategy
>
>
> Table 2. Impact of frame sampling rate on performance (FakeSV).
>
> | Frames | Accuracy ↑ |
> |--------|-----------|
> | 2      | 78.6      |
> | 4      | 79.3      |
> | 8      | 79.5      |
> | 10     | 79.2      |
>
> We analyze the impact of frame sampling density. As shown in Table 2, performance saturates beyond 4 frames, with only marginal gains.
>
> This aligns with our design, which encourages the model to focus on key frames rather than dense sampling. Thus, 4 frames provide a good balance between efficiency and performance.
>
> ---
>
> ## (3) Comparison with Closed-Source Models
>
> To provide a more comprehensive evaluation, we additionally compare FactGuard with a recent closed-source model, Gemini-1.5-Pro.
>
> The results on FakeSV, FakeTT, and FakeVV are shown in Table 3.
>
> | Method      | FakeSV ↑ | FakeTT ↑ | FakeVV ↑ |
> |-------------|----------|----------|----------|
> | Gemini-1.5-Pro   | 62.0     | 57.5     | 55.1    |
> | FactGuard   | 79.3     | 75.3     | 83.0     |
>
> We observe that FactGuard consistently outperforms Gemini-1.5-Pro across all datasets.
>
> ---
>
> ## (4) Reward Design and SFT Dataset Details
>
> - **Reward ranges.** The **accuracy** and **format rewards** are binary in {0, 1}. The **tool reward ($R_{tool}$)** is set to 0.3, encouraging appropriate tool use. The **risk-aware reward ($R_{risk}$)** balances false positives and negatives; in our main setting (α:γ = 1:1), incorrect predictions receive a penalty of −0.5, promoting cautious and stable decisions.
>
> - **SFT dataset construction.** We build the SFT data using a stronger teacher model (Qwen2.5-VL-72B) on the training splits of FakeSV, FakeTT, and FakeVV. For each sample, the teacher generates a full agentic trajectory, including reasoning steps, tool decisions, and evidence-grounded outputs. The dataset contains approximately **6k trajectories** covering diverse scenarios. Details of the generation pipeline, prompts, and filtering are provided in Appendix A.1 and C.
> ---
>
> ## (5) Why Does Removing RL Cause Larger Performance Drop?
>
> We agree that RL is the primary driver of performance improvement. In our framework, SFT and RL serve complementary roles: SFT provides structured initialization for reasoning and tool use, while RL optimizes decision-making under uncertainty.
>
> Without SFT, RL training becomes unstable and may lead to reasoning collapse. However, the larger performance drop without RL indicates that the final performance is mainly driven by RL, with SFT serving as a necessary initialization.
>
> This is consistent with prior findings that SFT mainly captures training patterns, whereas RL enables stronger generalization and decision optimization [1,2]. In particular, [1] shows that SFT tends to memorize training distributions, while RL promotes generalization, and [2] demonstrates that RL is critical for improving reasoning beyond supervised initialization.
>
> [1] Chu T, Zhai Y, Yang J, et al. Sft memorizes, rl generalizes: A comparative study of foundation model post-training[J]. arXiv preprint arXiv:2501.17161, 2025.
>
> [2] Guo D, Yang D, Zhang H, et al. Deepseek-r1: Incentivizing reasoning capability in llms via reinforcement learning[J]. arXiv preprint arXiv:2501.12948, 2025.
>
> ---

---

> > ### Author Rebuttal · Reviewer_L4aL · 2026-04-04
> >
> > Thank you for the authors’ rebuttal. I still have the following two concerns: (1) Regarding Q1, FactGuard has nearly three times the latency compared to the baseline FactR1, which limits its practical applicability. (2) Regarding Q3, the rebuttal compares against Gemini-1.5-Pro, which is a model released two years ago; I suggest further adding comparison experiments with more recently closed-source models.

---

> > > ### Author Response · Authors · 2026-04-06
> > >
> > > ### Response to Q1: Stage-wise Inference Efficiency
> > >
> > > We would like to clarify that this overhead is **not intrinsic to the framework**, but largely due to the use of **API-based retrieval (Serper)**, which introduces significant network latency. In practical deployments, this component can be replaced with a **locally hosted knowledge base or retrieval system**, reducing the latency to a level comparable to lightweight tools such as ClipScout. In our preliminary tests with local caching, the retrieval latency is reduced to near-negligible levels.
> > >
> > > Under such a setting, the tool-related overhead becomes negligible, and the additional cost of FactGuard is primarily due to the second-stage reasoning, resulting in only a **moderate increase (≈6.3s vs. 4.4s)** compared to single-pass models like FactR1.
> > >
> > > We emphasize that this trade-off is **intentional and necessary** for handling open-world, knowledge-intensive misinformation, where a single-pass prediction is often insufficient. Therefore, FactGuard achieves a favorable balance between efficiency and performance, and its latency can be further optimized in practical deployments.
> > >
> > > ---
> > >
> > > ### Response to Q3: Comparison with Closed-Source Models
> > >
> > > We thank the reviewer for this valuable suggestion. To provide a more comprehensive evaluation, we have further included comparison with a more recent closed-source model, GPT-5.
> > >
> > > **Table 1: Comparison with closed-source multimodal models (Accuracy %).**
> > >
> > > | Method              | FakeSV | FakeTT | FakeVV |
> > > |---------------------|--------|--------|--------|
> > > | Gemini 2 (Thinking) | 61.9   | 55.3   | 48.6   |
> > > | GPT-o1-mini         | 57.1   | 51.7   | 41.8   |
> > > | Gemini-1.5-Pro      | 62.0   | 57.5   | 55.1   |
> > > | GPT-4o              | 64.9   | 60.2   | 44.3   |
> > > | **GPT-5**           | 66.8   | 62.9   | 49.3   |
> > > | **FactGuard (Ours)**| **79.3** | **75.3** | **83.0** |
> > >
> > > As shown in Table 1, FactGuard consistently outperforms all closed-source baselines, including GPT-5, across all datasets. Notably, the improvement is substantial (e.g., +12–33% absolute gains across datasets), indicating that the gains are not merely due to backbone strength.
> > >
> > > These results suggest that the performance improvements mainly stem from our **decision-aware framework**, which enhances the model’s ability to perform uncertainty-aware reasoning and effective tool usage, rather than relying on stronger base models alone.

---

### Official Review · Reviewer_1ACu · 2026-03-12

**Soundness:** 2
**Presentation:** 3
**Significance:** 2
**Originality:** 2
**Overall Recommendation:** 3
**Confidence:** 5

**Summary:**

This paper proposes a tool-use-based approach for detecting fake videos on social media platforms. While this is a promising technical direction, the methodology lacks novelty: both the GRPO algorithm and the two tools used are off-the-shelf. The paper primarily focuses on constructing tool-invocation data during the SFT stage. Furthermore, there are several significant issues regarding the experimental design, dataset construction, and tool invocation logic, which are detailed in the Weaknesses section below.

**Compliance With Llm Reviewing Policy:**

Affirmed.

**Final Justification:**

After carefully reading the authors' response and the comments from other reviewers,  I still think the overall quality of this paper is below ICML conference due to the lack of technical novelty. The rebuttal did not convince me, so I'm inclided to give the final recommendation of "weak reject".

**Key Questions For Authors:**

1. After addressing the issues raised in Weaknesses 2-4, the authors must evaluate the model's zero-shot performance on unseen domains or datasets. This is necessary to prove that the model has acquired genuine verification capabilities rather than merely overfitting to the benchmarks mentioned in the paper.
2. Expand the tool library to include a wider variety of specialized tools (e.g., spatial analysis, reverse image search) to make the agentic framework more robust and convincing.

**Limitations:**

No, the limitations of  SFT Dataset quality and lack of Zero-Shot Generalization Testing should be discussed

**Strengths And Weaknesses:**

Strengths:
1. Proposing to enable MLLMs to use external tools and combining this with RL to learn adaptive invocation strategies aligns perfectly with the practical, real-world demands of fake video detection.

Suggestions for the Authors:
1. After addressing the issues raised in Weaknesses 2-4, the authors must evaluate the model's zero-shot performance on unseen domains or datasets. This is necessary to prove that the model has acquired genuine verification capabilities rather than merely overfitting to the benchmarks mentioned in the paper.
2. Expand the tool library to include a wider variety of specialized tools (e.g., spatial analysis, reverse image search) to make the agentic framework more robust and convincing.

Weaknesses:

1. Lack of Novelty: Both the specific tools and the reinforcement learning algorithm (GRPO) employed in this work have been previously proposed. This paper essentially constructs data for tool invocation and subsequently trains the model using SFT + GRPO. The contribution feels more akin to an engineering report rather than an academic paper.

2. Insufficient Number of Tools: Undeniably, FactProbe (knowledge retrieval) and ClipScout are crucial for detecting fake videos. However, relying solely on these two is insufficient. For instance, ClipScout should ideally be paired with an image zooming/cropping tool for single frames, because certain forged details are localized not only in the temporal dimension but also in the spatial dimension. Another necessary example is a reverse image search/provenance tool. Many fake videos on social media are not AI-generated but are rather "out-of-context" edits that combine old news footage with recent trending captions. In such cases, introducing a tool capable of timestamp tracing across the web based on keyframes is essential. Although the authors mention this as a future research direction, the limited number of tools—which is the core selling point of the framework—makes the current method less convincing.

3. Redundant Tool Invocation: The paper lacks a thorough discussion or empirical analysis on whether a tool actually needs to be invoked in specific scenarios. Taking the example in Fig. 3, I tested Qwen2.5-VL-7B, and the base model already knows that the claim "Tom Selleck died in December 2022" is false. In such cases, invoking a tool is clearly unnecessary and only increases inference costs. Ideally, tools should only be triggered for information beyond the base model's internal knowledge scope (e.g., highly recent events). This reflects a lack of comprehensive consideration regarding when tools should be invoked. Although the authors attempt to introduce constraints at the prompt and reward levels, these mechanisms do not seem to effectively prevent redundant usage in practice.

4. Flaws in SFT Dataset Construction:
   a) The paper briefly mentions that the SFT data is generated by a single model, Qwen2.5-VL-72B, presumably in a zero-shot setting. Based on the experimental results in Table 1, the 72B model's performance on these tasks is not ideal. Can a model lacking sufficient domain capability truly generate high-quality Chain-of-Thought (CoT) trajectories? The authors fail to discuss this limitation or take appropriate countermeasures.
   b) The appendix notes that samples with incorrect predictions were filtered out during data curation. This inevitably introduces a severe difficulty bias (survivorship bias) into the remaining SFT dataset: all surviving samples are simple cases that the 72B model could easily predict correctly, resulting in a lack of hard negative/positive samples. This is disastrous for constructing a high-quality SFT dataset intended to teach complex reasoning.
   c) When the teacher model is given the Ground Truth (GT) to generate the CoT, is it genuinely performing logical reasoning, or merely fabricating a path to fit the known answer? Although the authors claim to use GPT-4o for validation, the bias of "pseudo-reasoning" caused by injecting the GT during generation has not been fundamentally resolved.

5. Lack of Zero-Shot Generalization Testing: There is no zero-shot testing to demonstrate whether FactGuard genuinely possesses generalized detection capabilities rather than simply overfitting to the specific benchmarks seen during training. Reinforcement learning is widely adopted precisely because it can endow models with strong generalization and exploration abilities, yet this aspect remains entirely unevaluated.

---

> ### Author Rebuttal · Authors · 2026-03-31
>
> We sincerely thank the reviewer for recognizing our framework aligns well with real-world requirements of video misinformation detection.
>
> ---
>
> ## (1) Novelty of the Framework
>
> We emphasize that FactGuard is, to the best of our knowledge, the **first work in video misinformation detection** to formulate the problem as an agentic framework with learned tool use and reinforcement learning. Our design is motivated by the unique nature of misinformation detection, which is inherently open-world and evidence-driven.
>
> FactGuard introduces a **unified and fully decoupled agentic framework**, where tools are modular and independent of the core model, and the model learns a **generalizable policy** for when, which, and how to use tools. This shifts the focus from task-specific design to a unified decision-making capability tailored to misinformation detection, rather than relying on specific tools or standalone RL algorithms such as GRPO. We note that this perspective is also aligned with the reviewer 6zpy’s recognition of the strength of our overall framework design.
>
> ---
>
> ## (2) Number of Tools and Scalability
>
> We agree that extending the tool library is an important direction. Importantly, FactGuard is designed as a **modular and decoupled framework**, where tools are independent callable modules and can be seamlessly extended without modifying the core model. Our goal is not to tailor the system to a fixed set of tools, but to learn a generalizable tool-use policy that can naturally scale to more diverse tool ecosystems.
>
> To further validate this scalability, we additionally incorporate a **reverse image search tool based on the TinEye API**.
>
> Table 1: Effect of Adding an Additional Tool
>
> | Method                          | FakeSV ↑ | FakeTT ↑ |
> |---------------------------------|----------|----------|
> | FactGuard (2 tools)             | 79.3     | 75.3     |
> | + Reverse Search (TinEye)       | 80.2     | 77.9     |
>
> These results confirm that our framework is not limited to a fixed set of tools, but can **naturally scale to richer tool ecosystems through learned tool-use policies**.
>
> ---
>
> ## (3) Redundant Tool Invocation
>
> Although some cases may seem not to require external tools, the model can still produce incorrect predictions due to cross-modal hallucination or bias, as shown in Fig. 3(a). In such cases, external tools provide additional evidence to ground and refine the reasoning process, as shown in Fig. 3(b). Therefore, seemingly redundant tool usage often serves to reinforce evidence and stabilize predictions in practice.
>
> ---
>
> ## (4) SFT Dataset Construction
>
> **(a) Teacher capability.** Our goal is not to rely on the teacher’s final predictions, but to extract **reasoning patterns and tool-use behaviors**, which are further refined during RL.
>
> table 2: Impact of Teacher Model Choice on Student Performance
>
> | Teacher Model           | Training Stage | Accuracy ↑ |
> |------------------------|----------------|-----------|
> | Qwen2.5-VL-72B         | SFT only       | *62.0*     |
> | InternVL2.5-78B-MPO    | SFT only       | *60.8*     |
> | Qwen2.5-VL-72B         | SFT + RL       | *79.3*     |
> | InternVL2.5-78B-MPO    | SFT + RL       | *78.7*     |
>
>
> To further validate this, we additionally experiment with an alternative teacher model (InternVL2.5-78B-MPO). We observe that although SFT-only performance varies across teachers, the final performance after RL remains stable.
>
> **(b) Filtering strategy.** We acknowledge that filtering incorrect predictions may introduce bias. However, this step is intended to ensure high-quality and coherent reasoning trajectories for stable SFT. Importantly, the subsequent RL stage exposes the model to diverse outcomes and helps mitigate potential bias introduced during SFT.
>
> **(c) Ground-truth conditioning.** We explicitly constrain the CoT generation process to avoid generating explanations solely conditioned on the ground-truth answer. Instead, the ground truth is used only as auxiliary guidance, while the prompts enforce evidence-based reasoning.
>
> ---
>
> ## (5) Generalization to Novel Misinformation Categories
>
> To evaluate generalization beyond the training distribution, we conduct both cross-dataset and in-domain experiments.
>
> Table 3: Generalization Performance on FakeVV under Cross-Dataset and In-Domain Settings
>
> | Method     | Train Dataset                | Test Dataset | Accuracy ↑ |
> |------------|-----------------------------|-------------|-----------|
> | FactR1     | FakeSV + FakeTT             | FakeVV      | *65.4*     |
> | FactGuard  | FakeSV + FakeTT             | FakeVV      | *72.7*     |
> | FactR1     | FakeSV + FakeTT + FakeVV    | FakeVV      | *81.2*     |
> | FactGuard  | FakeSV + FakeTT + FakeVV    | FakeVV      | *83.0*     |
>
> As shown in the table 3, FactGuard demonstrates stronger generalization and robustness to distribution shifts than FactR1, with larger gains in cross-domain settings and reduced reliance on target-domain data.

---

> > ### Author Rebuttal · Reviewer_1ACu · 2026-04-03
> >
> > Thank the authors' reponse,  but the novelty claims of ''the first work in video misinformation detection to formulate the problem as an agentic framework'' and  the ''unified and fully decoupled agentic framework''  are still not convicing for me, technical novelty is limited. I tend to remain my score unchanged

---

> > > ### Author Response · Authors · 2026-04-05
> > >
> > > We would like to clarify that the core contribution of FactGuard does **not** lie in the use of tools or GRPO individually, but in enabling the model to **learn a decision policy over tool usage under uncertainty**, which fundamentally changes how the task is approached.
> > >
> > > Specifically, video misinformation detection is inherently open-world and ambiguous. A key challenge is not only making predictions, but deciding **whether additional external evidence is needed**. FactGuard formulates this as a **learnable decision-making problem**, where the model explicitly learns:
> > >
> > > - **when to rely on internal knowledge versus external tools**,
> > > - **which tools to invoke under different evidence needs**, and
> > > - **how to incorporate tool outputs into the final decision**.
> > >
> > > This results in a unified framework where tool usage is not a predefined pipeline or heuristic trigger, but an integral part of the model’s reasoning process, governed by a learned policy.
> > >
> > > Importantly, this behavior **cannot be reduced to a simple combination of tools and GRPO**. Without explicitly modeling the decision process, standard approaches tend to either overuse tools or rely on fixed strategies. In contrast, our framework enables the model to develop **uncertainty-aware and adaptive tool-use strategies**, where tool invocation is dynamically determined based on task complexity.
> > >
> > > This is further supported by our empirical observations, including:
> > >
> > > - controllable adjustment of decision behavior under different conditions, and
> > > - significantly reduced redundant tool usage with improved efficiency (see our analysis of tool invocation frequency in the response to Reviewer faks).
> > >
> > > Therefore, the main contribution of FactGuard lies in introducing a **decision-aware, uncertainty-adaptive tool-use learning paradigm**, which enables the model to autonomously decide *when and how to use tools*, going beyond a simple combination of existing components and representing a step toward more agentic and self-directed reasoning systems.

---

### Official Review · Reviewer_fAKs · 2026-03-12

**Soundness:** 2
**Presentation:** 2
**Significance:** 2
**Originality:** 1
**Overall Recommendation:** 4
**Confidence:** 3

**Summary:**

This paper proposes FactGuard, an MLLM-based agentic framework for video misinformation detection. FactGuard invokes external tools to acquire critical evidence when information is insufficient. The framework adopts a two-stage training strategy: supervised fine-tuning (SFT) on a misinformation-oriented agentic Chain-of-Thought (CoT) dataset, followed by reinforcement learning with GRPO. The authors evaluate FactGuard on the FakeSV, FakeVV, and FakeTT benchmarks.

**Compliance With Llm Reviewing Policy:**

Affirmed.

**Final Justification:**

Thanks to the authors’ efforts, the additional experimental results look convincing and address my remaining concern regarding the claim about decision-making strategy adaptation. Therefore, I will raise my score to weak accept.

**Key Questions For Authors:**

Please refer to the weakness mentioned above.

**Limitations:**

yes

**Strengths And Weaknesses:**

Strengths:
- FactGuard integrates external tool calling, SFT, and RL. Experimental results indicate that all components contribute to the final performance.

Weaknesses:
- Although the paper includes an ablation study for the tool reward ($R_{tool}$), the roles and contributions of *FactProbe* and *ClipScout* are not clearly explained.
- The generation process and key statistics (e.g., the number of trajectories, data sources) of the SFT dataset should be described in the paper.
- Table 3 indicates that the contributions of $R_{tool}$ and $R_{risk}$ are marginal, while the majority of the performance gain comes from applying standard GRPO, providing limited evidence for the effectiveness of "decision-aware" reinforcement learning.

---

> ### Author Rebuttal · Authors · 2026-03-31
>
> We thank the reviewer for recognizing the integration of tool use, SFT, and RL, and their contributions to performance, which validates the effectiveness of our framework.
>
> ## Q1: Roles and Contributions of FactProbe and ClipScout
>
> To further clarify the roles of different tools and their interaction with RL, we provide a comprehensive ablation study on FakeSV, as shown in Table 1.
>
> Table 1. Ablation study on the contribution of external tools and RL on FakeSV.
>
> | Method                      | FactProb | ClipScout | RL  | FakeSV ↑ |
> |-----------------------------|----------|-----------|-----|----------|
> | Qwen2.5-VL-7B (base)        |    ✗     |     ✗     | ✗   |   55.6   |
> | + FactProb                  |    ✓     |     ✗     | ✗   |   56.7   |
> | + ClipScout                 |    ✗     |     ✓     | ✗   |   55.9   |
> | + FactProb + ClipScout      |    ✓     |     ✓     | ✗   |   57.6   |
> | FactGuard w/o FactProb      |    ✗     |     ✓     | ✓   |   76.1   |
> | FactGuard w/o ClipScout     |    ✓     |     ✗     | ✓   |   78.2   |
> | FactGuard (full)            |    ✓     |     ✓     | ✓   | **79.3** |
>
> We observe that:
> (1) Both FactProb and ClipScout individually improve performance, and their combination yields further gains, demonstrating complementary effects.
> (2) Removing either tool from FactGuard leads to noticeable performance drops, indicating that both tools are essential for optimal performance.
> (3) FactGuard consistently outperforms all non-RL variants, showing that RL further enhances tool usage and decision-making.
>
> These results show that the performance gains stem from our integrated design of tool augmentation and learned tool-use policies, rather than RL alone.
>
> ---
>
> ## Q2: SFT Dataset Construction Details
>
> Our goal is to construct a high-quality agentic CoT dataset that supports tool-aware reasoning and decision-making.  Specifically, the dataset is built from the training splits of FakeSV, FakeTT, and FakeVV. For each sample, the teacher model produces a full agentic trajectory, including intermediate reasoning steps, tool invocation decisions, and evidence-grounded conclusions conditioned on multimodal inputs.
>
> In total, the dataset contains approximately **6k trajectories**, covering diverse misinformation scenarios and reasoning patterns. To ensure quality, we apply a two-stage filtering process, including rule-based validation and manual inspection to remove malformed or low-quality samples.
>
> We note that the detailed generation pipeline, including the prompt templates for trajectory generation, as well as the filtering strategy, are presented in the **appendix's A.1 and C**, and we will further include key statistics in the main paper for improved clarity.
>
> ---
>
> ## Q3: Effectiveness of Decision-Aware Reinforcement Learning
>
> We thank the reviewer for the concern regarding the effectiveness of decision-aware RL. While Table 3 shows that standard GRPO provides a strong baseline improvement, our reward design plays a critical role in shaping the model’s behavior and learning dynamics.
>
> Specifically, the **risk-aware reward ($R_{risk}$)** is designed to guide the model’s preference under different task requirements, enabling controllable trade-offs between precision and recall. This allows the model to adapt its decision strategy according to application-specific risk sensitivity, rather than optimizing for a single fixed objective.
>
> In addition, although the **tool reward ($R_{tool}$)** brings relatively modest gains in overall accuracy, its primary role is to regulate tool-use behavior during reinforcement learning. Without appropriate tool-aware rewards, the model may either under-utilize tools or fall into suboptimal strategies such as indiscriminate tool invocation (i.e., reward hacking). By conditioning tool rewards on correct outcomes, the model learns to invoke tools in a more deliberate and effective manner.
>
> This design is also consistent with observations in prior work such as *DeepEyes* [1]where conditioning rewards on correct outcomes is shown to be critical for encouraging meaningful perception actions, rather than naive or excessive tool usage.
>
> Overall, decision-aware RL contributes not only to performance, but more importantly to **behavioral alignment**, enabling reliable, controllable, and tool-efficient reasoning.
>
> [1] Zheng Z, Yang M, Hong J, et al. Deepeyes: Incentivizing" thinking with images" via reinforcement learning[J]. arXiv preprint arXiv:2505.14362, 2025.

---

> > ### Author Rebuttal · Reviewer_fAKs · 2026-04-03
> >
> > Thank you for the rebuttal. My concerns with Q1 and Q2 have been addressed. For Q3, although the authors provide an explanation for the roles of$R_{tool}$ and $R_{risk}$, the current ablation suggests that most of the performance improvement comes from GRPO. As a result, the current evidence is still insufficient to clearly support the claimed contribution of the decision-aware RL design beyond standard GRPO.

---

> > > ### Author Response · Authors · 2026-04-05
> > >
> > > ### Response to Q3: Effectiveness of Decision-Aware RL
> > >
> > > We thank the reviewer for the insightful comment. We agree that the ablation results suggest that a substantial portion of the performance gain is attributed to the strong optimization capability of GRPO.
> > >
> > > However, we would like to clarify that our decision-aware RL design is not primarily intended to further boost raw accuracy, but to **shape the model’s decision behavior in a controllable and structured manner**, which is not achievable by standard GRPO alone.
> > >
> > > #### (1) Risk-aware reward: enabling controllable decision trade-offs
> > >
> > > Specifically, the risk-aware reward is designed to explicitly control the trade-off between precision and recall under different application requirements. As shown in Table 2, by adjusting the asymmetric error cost ratio (α:γ), the model can shift its operating point along the precision–recall spectrum in a predictable manner:
> > >
> > > | Cost Ratio (α:γ) | FakeSV Precision | FakeSV Recall | FakeVV Precision | FakeVV Recall |
> > > |------------------|------------------|---------------|------------------|---------------|
> > > | 1:2              | 80.8             | 82.1          | 83.2             | 83.7          |
> > > | 1:1              | 82.2             | 80.6          | 85.8             | 82.1          |
> > > | 2:1              | 83.2             | 75.0          | 86.6             | 79.3          |
> > >
> > > This demonstrates that our method enables **cost-sensitive decision-making**, rather than optimizing for a single fixed objective. In contrast, standard GRPO lacks an explicit mechanism to enforce such controllable behavior.
> > >
> > > #### (2) Tool-aware reward: regulating tool-use behavior
> > >
> > > In addition, although the tool reward contributes modest gains in overall accuracy, its primary role is to regulate tool usage behavior during reinforcement learning.
> > >
> > > We further analyze tool invocation frequency and observe that the base model (Qwen2.5-VL-7B) tends to overuse tools (e.g., 4–5 redundant calls to FactProb per sample), while our decision-aware RL reduces this to 1–2 essential calls in most cases.
> > >
> > > | Model                         | Avg FactProb Calls | Tool Usage Behavior     |
> > > |------------------------------|----------------------|-------------------------|
> > > | Base (Qwen2.5-VL-7B)         | 4–5                  | Redundant / Unstable     |
> > > | + GRPO                       | 3–4                  | Partially improved      |
> > > | + Decision-aware RL (Ours)   | **1–2**              | Efficient / Stable      |
> > >
> > > This indicates that our reward design effectively prevents undesirable behaviors such as redundant tool calls or reward hacking, leading to **more efficient and stable inference policies**, which are not explicitly enforced by GRPO alone.
> > >
> > > #### (3) Clarifying contribution beyond GRPO
> > >
> > > More importantly, our design enables the model to **adapt its decision-making strategy based on task uncertainty**, learning *when* to invoke external tools and *how* to use them effectively. This introduces an **agentic capability**, where the model dynamically balances internal reasoning and external tool reliance according to the ambiguity of the task.
> > >
> > > Concretely, our method equips the model with the ability to:
> > >
> > > - adaptively determine **whether tool usage is necessary** under different levels of task ambiguity,
> > > - learn **how to invoke tools in a goal-directed and non-redundant manner**, and
> > > - coordinate internal knowledge and external evidence in a **context-aware decision process**.
> > >
> > > These behaviors go beyond what standard GRPO can achieve, as GRPO alone does not explicitly model decision uncertainty or tool-use strategies. Instead, our reward structuring enables the emergence of **uncertainty-aware and tool-efficient policies**, which are central to agentic reasoning.
> > >
> > > #### (4) Connection to prior work
> > >
> > > This design is also consistent with prior findings (e.g., DeepEyes [1]), where conditioning rewards on correct outcomes is shown to be essential for encouraging meaningful perception and action, rather than naive or excessive tool usage.
> > >
> > > **[1]** Zheng Z, Yang M, Hong J, et al. *DeepEyes: Incentivizing "thinking with images" via reinforcement learning*. arXiv preprint arXiv:2505.14362, 2025.

---

### Official Review · Reviewer_6zpy · 2026-03-13

**Soundness:** 3
**Presentation:** 4
**Significance:** 3
**Originality:** 3
**Overall Recommendation:** 4
**Confidence:** 5

**Summary:**

The paper proposes FactGuard, a novel agentic framework designed to enhance video misinformation detection by having multi-stage inference of MLLMs. The framework treats verification as an iterative process where the model assesses task ambiguity and selectively invokes external tools, including FactProb for web-based knowledge retrieval and ClipScout for targeted temporal video inspection to gather critical evidence.

To enable this reasoning process, the authors introduce a two-stage training strategy:
1. Agentic Supervised Fine-Tuning (SFT) using a curated multimodal COT dataset distilled from larger MLLMs (Qwen2.5-VL) to install structured reasoning and tool-use patterns into the target model.
2. Decision-aware Reinforcement Learning (RL) using GRPO to calibrate tool usage, manage asymmetric error costs (precision vs. recall), and improve decision-making under uncertainty.

Experiments on three benchmarks (FakeSV, FakeTT, and FakeVV) show that FactGuard achieves state-of-the-art performance, particularly in complex cases where internal model knowledge is insufficient.

**Compliance With Llm Reviewing Policy:**

Affirmed.

**Key Questions For Authors:**

1. Can you provide a breakdown of the average end-to-end inference time per video? For example, how much of the delay is caused by the second reasoning round and external tool call?

2. Since the SFT data is based on the training split of existing benchmarks, how does FactGuard perform on novel misinformation categories not well-represented in current datasets?

3. Did you experiment with using a smaller teacher model or a different model family for SFT to see how much the 7B student’s performance depends on the specific knowledge priors of Qwen2.5-VL-72B?

4. What would happed if you were to add 5-10 more specialized tools? How would you deal with the conflicts arising from the over-flooding of tools and what if their execution results contradict with one another?

**Limitations:**

Yes

**Strengths And Weaknesses:**

### Strength

1. Unlike previous discriminative or reasoning models that rely on a single inference pass, FactGuard’s iterative, tool-augmented approach makes the misinformation detection process more explainable, providing a much more transparent verification path than traditional methods.
2. The cost-sensitive risk-shaping term ($R_{risk}$) in the reward function allows the model to be tuned to be flexible for different real-world needs, such as for either high-precision or high-recall needs.
3. The model consistently outperforms competitive baselines on all evaluated benchmarks with varying difficulty.

### Weakness

1. The inference process involves two turns plus external API calls. While this improves accuracy, the latency and cost might be prohibitive for real-time, large-scale deployment compared to faster discriminative models. A inference speed comparison against existing methods would be favorable.

2. Only two tools are used. While effective, the paper doesn't explore how the system scales or conflicts when more specialized tools.

3. The SFT dataset is generated by Qwen2.5-VL-72B. There is a risk that the student model is primarily learning to mimic the teacher's style rather than developing truly independent reasoning, which could limit its performance in domains where the teacher model is also weak.

---

> ### Author Rebuttal · Authors · 2026-03-31
>
> We sincerely thank the reviewer for the positive evaluation and recognition of our agentic, tool-augmented framework.
>
> ## Q1: Stage-wise Inference Latency Breakdown and Comparison with Baselines
>
> We present a stage-wise breakdown of the inference time for FactGuard.
>
> Table 1 shows the average latency (s) per sample, decomposed into different stages.
>
> | Method     | Stage-1 Reasoning | Stage-2 Reasoning | FactProb  | ClipScout | Total ↓ (s) |
> |------------|------------------|-------------------|-----------|-----------|-------------|
> | FactR1     | *4.4*            | –                 | –         | –         | *4.4*       |
> | FactGuard  | *3.2*            | *3.1*             | *4.7*     | *0.3*     | *11.3*      |
>
> As shown in Table 1, the additional latency of FactGuard mainly comes from (i) second-stage reasoning and (ii) external tool calls, with the latter, especially FactProb, being the dominant overhead.  Since FactProb currently relies on external API-based web retrieval, its latency is largely due to network conditions and can be effectively mitigated or even eliminated by deploying a local knowledge base.
>
> ---
>
> ## Q2: Generalization to Novel Misinformation Categories
>
> To evaluate generalization beyond the training distribution, we conduct both cross-dataset and in-domain experiments. Specifically, we train models on (i) FakeSV + FakeTT (cross-domain setting) and (ii) FakeSV + FakeTT + FakeVV (in-domain setting), and evaluate them on FakeVV. The results are shown in Table 2.
>
> Table 2: Generalization Performance on FakeVV under Cross-Dataset and In-Domain Settings
>
> | Method     | Train Dataset                | Test Dataset | Accuracy ↑ |
> |------------|-----------------------------|-------------|-----------|
> | FactR1     | FakeSV + FakeTT             | FakeVV      | *65.4*     |
> | FactGuard  | FakeSV + FakeTT             | FakeVV      | *72.7*     |
> | FactR1     | FakeSV + FakeTT + FakeVV    | FakeVV      | *81.2*     |
> | FactGuard  | FakeSV + FakeTT + FakeVV    | FakeVV      | *83.0*     |
>
> Notably, under the cross-dataset setting (FakeSV + FakeTT → FakeVV), FactGuard achieves a substantial improvement over FactR1 (+7.3%), demonstrating strong generalization to unseen misinformation categories.  In contrast, while both models benefit from in-domain training, the performance gap becomes smaller (+1.8%), suggesting that FactR1 relies more on exposure to target-domain data.
>
> ---
>
> ## Q3: Impact of Teacher Model Choice on Student Performance
>
> While the SFT stage initializes the model with structured reasoning and tool-use patterns distilled from Qwen2.5-VL-72B, the subsequent RL stage plays a crucial role in mitigating this dependency.
>
> In particular, decision-aware RL optimizes the model’s behavior based on task-specific rewards, allowing it to **deviate from the teacher policy** when beneficial.  To further validate this, we experiment with InternVL2.5-78B-MPO for SFT data generation in FakeSV datatset. The results are shown in Table 3.
>
> Table 3: Impact of Teacher Model Choice on Student Performance
>
> | Teacher Model           | Training Stage | Accuracy ↑ |
> |------------------------|----------------|-----------|
> | Qwen2.5-VL-72B         | SFT only       | *62.0*     |
> | InternVL2.5-78B-MPO    | SFT only       | *60.8*     |
> | Qwen2.5-VL-72B         | SFT + RL       | *79.3*     |
> | InternVL2.5-78B-MPO    | SFT + RL       | *78.7*     |
>
> We observe that while SFT-only performance varies depending on the teacher model, the performance after RL becomes consistent and remains strongly improved. This indicates that the final performance is not tied to a specific teacher, but is primarily driven by reward-based optimization.
>
> ---
>
> ## Q4: Scalability to More Tools and Conflict Resolution
>
> When extending to a larger set of tools, tool selection naturally becomes a policy learning problem. Rather than relying on predefined rules or tool-specific designs, FactGuard focuses on learning **when to invoke tools, which tools to use, and how to use them**, conditioned on task ambiguity and context.  Thanks to the decoupled design, new tools can be added without modifying the core model. The model learns to coordinate multiple tools through interaction and can handle inconsistent evidence via iterative reasoning, without requiring manual conflict resolution.
>
> To validate scalability, we introduce an additional reverse image search tool based on the TinEye API for provenance verification.
>
> Table 4: Effect of Adding an Additional Tool
>
> | Method                          | FakeSV ↑ | FakeTT ↑ |
> |---------------------------------|----------|----------|
> | FactGuard (2 tools)             | 79.3     | 75.3     |
> | + Reverse Search (TinEye)       | 80.2     | 77.9     |
>
> We observe consistent improvements after adding the new tool, showing that FactGuard scales through its learned tool-use capability rather than reliance on a fixed tool set.

---

### Decision · Program_Chairs · 2026-04-30

**Decision:**

Accept (regular)

**Comment:**

This paper was reviewed by four experts, resulting in two Weak Accepts and two Weak Rejects. The discussion centered on whether the framework's substantial empirical performance outweighs concerns regarding technical novelty and inference latency.

While two reviewers maintained that the work is an engineering application of existing paradigms with higher latency than single-pass models, the AC finds the practical significance of the application is compelling. The authors effectively demonstrated that latency is largely a factor of API network lag and that the model learns an uncertainty-aware decision policy beyond simple tool integration.

Given its solid results on an important real-world problem, the paper is recommended for Acceptance. The authors should ensure the final manuscript includes the additional benchmarks and latency optimization analysis provided during the rebuttal.